# ZSP1601, a novel pan-phosphodiesterase inhibitor for the treatment of NAFLD, A randomized, placebo-controlled phase Ib/IIa trial

Yue Hu [1], Haijun Li[2,3], Hong Zhang[1], Xiaoxin Chen[2], Jinjun Chen[4], Zhongyuan Xu[4], Hong You [5], Ruihua Dong[5], Yun Peng[2], Jing Li[2], Xiaojiao Li[1], Dandan Wu[1], Lei Zhang[6], Di Cao[5], He Jin[5], Dongdong Qiu[6], Aruhan Yang[1], Jinfeng Lou[1], Xiaoxue Zhu [1]✉, Junqi Niu [7]✉ & Yanhua Ding [1]✉

Non-alcoholic fatty liver disease is a growing health burden with limited treatment options worldwide. Herein we report a randomized, double-blind, placebo-controlled, multiple-dose trial of a first-in-class pan-phosphodiesterase inhibitor ZSP1601 in 36 NAFLD patients (NCT04140123). There were three cohorts. Each cohort included twelve patients, nine of whom received ZSP1601 50 mg once daily, 50 mg twice daily, or 100 mg twice daily, and three of whom received matching placebos for 28 days. The primary outcomes were the safety and tolerability of ZSP1601. A total of 27 (27/36, 75%) patients experienced at least one treatment-emergent adverse event (TEAE). Most TEAEs were mild to moderate. There was no Serious Adverse Event. Diarrhea, transiently elevated creatinine and adaptive headache were frequently reported adverse drug reaction. We conclude that ZSP1601 is well-tolerated and safe, showing effective improvement in liver chemistries, liver fat content and fibrosis in patients with NAFLD.

Nonalcoholic steatohepatitis (NASH), a progressive and severe subtype of nonalcoholic fatty liver disease (NAFLD), is characterized by hepatic steatosis and necroinflammation with hepatocyte injury, along with a rapid progression of liver fibrosis[1-3]. It is estimated that NASH affects 1.5–6.5% of the adult population worldwide, some of whom may subsequently develop serious liver diseases, such as cirrhosis, hepatocellular carcinoma (HCC), or even liver failure in need of liver transplantation[3-7]. Over the past few decades, the prevalence of NASH has been on the rise in both developing and developed countries, and the increase has occurred in parallel with the rise in metabolic syndrome, dyslipidemia, obesity, and type 2 diabetes mellitus (T2DM)[4,8,9]. Although NASH is a global health concern and is becoming a leading cause of end-stage liver disease, the medications for NASH are still limited[10]. Therefore, it is essential that effective NASH-specific therapies are developed.

A growing number of potential drugs with various mechanisms of action are undergoing clinical trials to assess their usefulness in the treatment of NASH[11-14]. These therapeutic interventions mainly target lipid metabolism, insulin signaling, inflammatory cytokine signaling, or pathways implicated in the pathogenesis of NASH-associated liver

[1]Phase I Clinical Research Center, First Hospital of Jilin University, Changchun, China. [2]Guangdong Raynovent Biotech Co., Ltd, Guangzhou, China. [3]Department of Anatomy and Neurobiology, School of Basic Medical Science, Central South University, Changsha, China. [4]Nafang Hospital, Nanfang Medical University, Guangzhou, China. [5]Beijing Friendship Hospital, Capital Medical University, Beijing, China. [6]Department of Radiology, First Hospital of Jilin University, Changchun, China. [7]Department of Hepatology, First Hospital of Jilin University, Changchun, China. ✉e-mail: zhuxiaoxue@jlu.edu.cn; junqiniu@jlu.edu.cn; dingyanh@jlu.edu.cn

fibrosis[13–16]. Among therapeutic agents in ongoing clinical trials for NASH, only a number of candidates have entered phase 3 clinical trials[13,15,17–19]. For instance, aramchol, a synthetic conjugate of cholic acid and arachidic acid, exerts its therapeutic effects in NASH through inhibition of stearoyl-Coenzyme A desaturase 1 (SCD-1), obeticholic acid (OCA), and resmetirom (MGL-3196). In addition, vitamin E treatment has also been investigated in phase 2 clinical trials for its usefulness in the treatment of NASH[20]. It has been shown to reduce steatosis and improve histological features of NASH in patients without concomitant diabetes or cirrhosis. In fact, the American Association for the Study of Liver Diseases (AASLD) has recommended the use of high-dosage vitamin E (800 IU daily) in patients with NASH without concomitant diabetes[21]. Despite these demonstrated benefits of vitamin E, other phase 2 trials have reported significant adverse effects of high-dose vitamin E, including an increased risk of bleeding as well as other adverse effects[21]. Thus, it appears unlikely that high-dose vitamin E will be further investigated in phase 3 clinical trials. Currently, saroglitazar, a peroxisome proliferator activated receptor (PPAR)-α/γ agonist, is approved drug for the treatment of non-cirrhotic NASH in India[22,23]. At present, numerous potential drugs with various mechanisms of action are in development as potential treatments for NASH.

ZSP1601 is a first-in-class pan-phosphodiesterase (pan-PDE) inhibitor specially designed for NASH and developed by Guangdong Raynovent Biotech Co., Ltd (Guangdong, Guangzhou, China). Previous in vitro and in vivo studies have shown that ZSP1601 exerted anti-inflammatory activities, with its highest inhibitory effect occurring on PDE2 that can increse the intracellular concentration of cAMP to suppress the synthesis of tumor necrosis factor alpha (TNF-α), a proinflammatory cytokine that induces inflammation and hepatocellular damage in NASH[24,25]. Notably, its half-maximal inhibitory concentration (IC50) was 2.82 μM, considerably lower than the IC50 of pentoxifylline (PTX, 34.0 μM), a proven inhibitor of phosphodiesterase with anti-inflammatory properties[26,27]. In addition to PDE2, ZSP1601 has inhibitory effects on other subsets of PDE, including PDE1A, PDE4B, and PDE5A. In preclinical studies, ZSP1601 demonstrated a greater inhibitory effect on PDE2 in comparison with PTX[27]. Recently, a phase 1 clinical trial of healthy Chinese individuals investigated the pharmacokinetics (PK), safety, and tolerability of ZSP1601, including single dose (25–350 mg), multiple doses (50–100 mg QD), and a two-period crossover food effect study (100 mg)[27]. The findings in a phase 1 clinical trial of healthy Chinese individuals indicated that ZSP1601 was rapidly absorbed, with maximum plasma concentrations ($C_{max}$) reached at 1.25 to 2.50 h (median $T_{max}$) and safe. These results are promising. Thus, we conducted this randomized, double-blind, placebo-controlled, multiple-dose, phase Ib/IIa clinical trial, aiming to assess the safety, tolerability, pharmacokinetics, and efficacy of ZSP1601 in patients with NAFLD. Given that the drug was first used in NAFLD patients, safety and tolerability are the primary outcomes, while efficacy and pharmacokinetics are the secondary outcomes. Recently, magnetic resonance imaging-derived proton density fat fraction (MRI-PDFF), a noninvasive measurement of the amount of hepatic fat, has been validated as an accurate biomarker of liver steatosis[28,29]. In previous clinical trials for NASH, MRI-PDFF had been applied to assess the efficacy of drugs in reducing liver fat[30–32]. In this study, MRI-PDFF was used to quantify liver fat content (LFC) in study subjects before and after treatment completion. FibroScan examinations with transient elastography technology were used to assess liver fibrosis based on liver stiffness measurements (LSM) and liver steatosis using controlled attenuation parameters (CAP). A FibroScan-aspartate aminotransferase (FAST) score was used for evaluation of NAFLD. Here, we show that ZSP1601 is well-tolerated and significantly reduces liver chemistries and liver fat content, and improves fibrosis in patients with NAFLD after 28 days of continuous treatment. These findings highlight the therapeutic potential of ZSP1601 in the treatment of NAFLD in this phase Ib/IIa study. These results warrant further clinical trials to evaluate the safety, efficacy and pharmacokinetics of ZSP1601 in the treatment of NAFLD.

## Results

### Demographic and clinical characteristics of study subjects

Among the 95 NAFLD patients who were screened, 37 patients were randomized, and 36 received placebo or ZSP1601 (50 mg QD, 50 mg BID, or 100 mg BID), and 36 (100%) completed the entire clinical study (Fig. 1). Of the 36 enrolled patients, the presence of NAFLD was confirmed in 2 patients (1 in the placebo group and 1 in the ZSP1601 50 mg BID group) histologically via liver biopsy, resulting in a NAS of 4 and 5, respectively. The remaining 34 patients were diagnosed based on clinical features of NAFLD All 36 patients who received treatment were included in an assessment of outcomes, while one randomized patient who did not receive treatment was excluded from all analysis of this study. The demographic and clinical characteristics of the study subjects at baseline are listed in Table 1. There was an imbalance in the distribution of age and sex among the study subjects at baseline between the ZSP1601 and placebo groups that was caused by randomization (Table 1). Males had significantly higher incidence and prevalence rate of NAFLD, consistent with literature reports[33,34]. Of the NAFLD patients, 15 had a history of other medical conditions, including hypertension as the most common condition other than NAFLD (1 in the ZSP1601 50 mg QD group, 2 in the 50 mg BID, and 2 in 100 mg BID group), followed by T2DM (1 each in the placebo, ZSP1601 50 mg QD, and 100 mg BID groups). Laboratory and clinical characteristics at baseline, including ALT, AST, LFC measured by MRI-PDFF, FibroScan parameters (LSM and CAP), and other selected characteristics, were comparable among the ZSP1601 and placebo groups (Table 1).

### Safety and tolerability

A total of 27 (27/36, 75%) patients experienced at least one AE. Most AEs were mild to moderate (Grades 1–2) based on the CTCAE, v5.0 (Table 2). There were no significant differences in the incidence rates of AEs between the ZSP1601 50 mg QD (77.8%), 50 mg BID (66.7%), 100 mg BID (88.9%), and placebo groups (66.7%). Adverse drug reactions (ADRs) occurred in 18 (18/36, 50%) patients. The overall incidence of ADRs was similar between groups except ZSP1601 100 mg BID (ZSP1601 50 mg QD, 44.4%, 50 mg BID, 44.4%, 100 mg BID, 77.8%, and placebo, 33.3%).

The most common ADRs in this study (with an incidence rate >5%) included diarrhea 16.7% (6/36), slightly and transiently elevated levels of creatinine (13.9%, 5/36), adaptive headache (13.9%, 5/36), indigestion (8.3%, 3/36), and abdominal pain (5.6%, 2/36). It was found that 6 patients (1 (11.11%) in the placebo group, 2 (22.2%) in the ZSP1601 50 mg BID group, and 3 (3/9, 33.3%) in the ZSP1601 100 mg QID group) experienced 8 AEs in need of symptomatic treatment, including diarrhea, supraventricular extrasystoles, chest pain, myalgia or muscle pain, headache, trauma, and nasopharyngitis. These AEs resolved completely after treatment. Four patients experienced grade 3 or 4 hypertriglyceridemia in the placebo group, while 2 patients experienced grade 3 hypertriglyceridemia in the ZSP1601 50 mg QD group. However, all AEs of grade ≥ 3 were neither drug-related nor SAE. No withdrawal due to AEs occurred. Collectively, ZSP1601 was safe and well-tolerated by patients in this study.

### Pharmacokinetics

Plasma pharmacokinetics after administration of ZSP1601 50 mg QD, 50 mg BID, or 100 mg BID were evaluated and the resulting pharmacokinetic parameters are summarized in Table 3 and Fig. 2. Determination of plasma concentrations of ZSP1601 and the metabolite M3−5 indicated that ZSP1601 was rapidly absorbed after oral administration of the drug. The resulting data showed that the plasma concentrations of ZSP1601 and M3−5 reached peak levels at 1−2.5 h and 3−8 h after

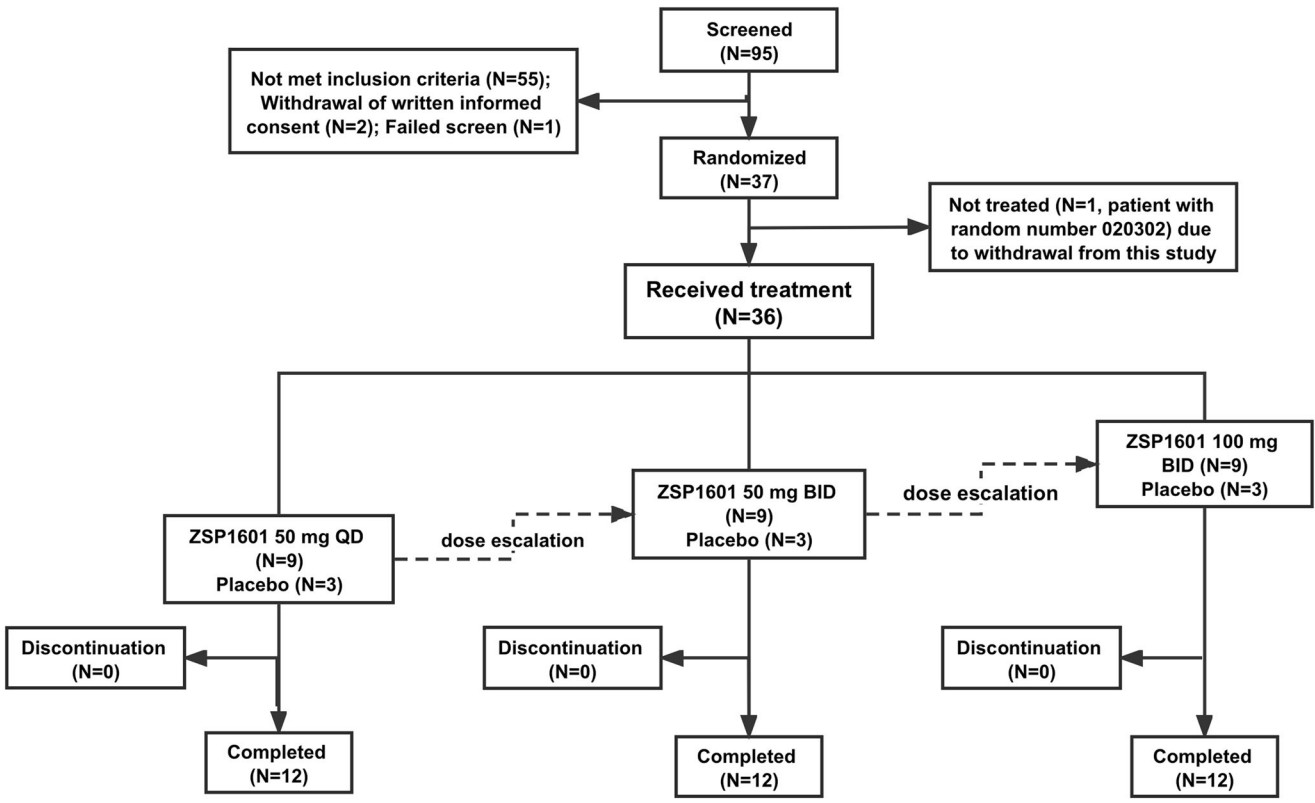

**Fig. 1 | Schematic diagram of patients selection, enrollment, and study design.** A total of 95 NASH patients were screened, of which 37 patients were randomized, and 36 NAFLD patients were enrolled in three cohorts. Each cohort included 12 patients, 9 of whom received ZSP1601 50 mg QD, 50 mg BID, or 100 mg BID, and 3 of whom received matching placebos for 28 days. All the 36 patients completed the clinical trial and were scheduled to receive a 14-day treatment-free follow-up.

dosing, respectively. The elimination half-life of ZSP1601 50 mg QD was 9.2 h. Plasma concentrations of ZSP1601 and M3–5 achieved steady state on day 3 and 4, respectively. The accumulation ratio of ZSP1601 and metabolite M3–5 plasma exposure ($R_{\text{acc\_AUC}}$) was 1.36–1.95 and 1.57–3.15. The key pharmacokinetic parameters of the metabolite M3–5 are presented in Table 3. The main pharmacokinetic parameters of the primary drug ZSP1601 and metabolite M3–5 exhibited a linear kinetic relationship over the dose range from 50 QD to 100 mg BID. Almost no drug accumulation was observed for ZSP1601, with slight accumulation of the metabolite M3–5. The pharmacokinetic characteristics were similar to those of healthy subjects.

**Efficacy**

We performed the analysis of efficacy of ZSP1601 in the 36 study subjects by comparing primary efficacy indicators (ALT, AST, GGT, ALP, and LFC) (Figs. 3–5) and secondary efficacy indicators (Supplementary Table 1) between groups of NAFLD patients.

ALT decreased with time in all ZSP1601 groups throughout treatment (Fig. 3a). After 28 days of treatment, the mean reduction from baseline in ALT was significantly greater than placebo for ZSP1601 50 mg BID and 100 mg BID groups [mean (SD), −39.03 (21.39) and −62.18 (26.63); $P = 0.0356$ and $P < 0.0001$] (Fig. 4a). The mean serum ALT levels were decreased by 14.06%, 20.93%, 35.69% and 49.03% in the placebo, ZSP1601 50 mg QD, 50 mg BID, and 100 mg BID groups, respectively. The proportion of NAFLD patients whose ALT levels returned to normal after 28 days of treatment were 0% (0/9), 0% (0/9), 22.22% (2/9), and 44.44% (4/9) in the placebo, ZSP1601 50 mg QD, 50 mg BID, and 100 mg BID groups, respectively. There was an apparent dose-dependence in patient responses. A reduction of ≥30% from the baseline ALT levels was found in 2 (2/9, 22.22%), 4 (4/9, 44.44%), 6 (6/9, 66.67%), and 8 (8/9, 88.89%) in the placebo, ZSP1601 50 mg QD, 50 mg BID, and 100 mg BID groups, respectively.

When stratified by BMI, there was a significantly larger reduction from baseline in serum ALT levels in the ZSP1601 100 mg BID groups compared with the placebo group ($P = 0.0125$) in patients with a BMI < 28 kg/m². In patients with ALT ≥ 100 U/L in men or ALT ≥ 80 U/L in women, significant reductions were observed in the 100 mg BID group versus the placebo group ($P = 0.0118$).

The time course of reduction in serum AST levels was similar to ALT (Fig. 3b). The relative reductions in AST levels from baseline were 12.06%, 16.87%, 28.76%, and 42.51% in the placebo, ZSP1601 50 mg QD, 50 mg BID, and 100 mg BID groups, respectively. Compared with patients who received placebo treatment, AST levels showed a significantly greater decrease from baseline in patients treated with ZSP1601 100 mg BID after the 28-day treatment [mean (SD), −30.41 (18.27); $P = 0.0041$] (Fig. 4b). The proportion of patients with reduced AST levels ≥ 30% from baseline was greater in the ZSP1601 100 mg BID group compared with the placebo group [difference between two groups, 44.44 (−0.65 to 71.00)].

Alterations in other liver chemistries, including serum GGT and ALP levels in placebo and ZSP1601 treatment groups are shown in Fig. 3c, d. After the treatment course, there were no significant differences in the reduction of serum ALP and GGT levels from baseline between the ZSP1601 and placebo groups (all $P > 0.05$).

Alterations of LFC from baseline immediately after completion of the 28-day treatment were examined by MRI-PDFF and the resulting data are illustrated in Fig. 5. The LS mean absolute reduction in LFC from baseline was dose-dependent [LS mean (95% CI), −2.49 (−5.52 to 0.54), −2.89 (−5.94 to 0.16), −4.98 (−7.62 to −2.33) and −5.42 (−8.64 to −2.19) in the placebo, ZSP1601 50 mg QD, 50 mg BID, and 100 mg BID groups, respectively; all $P > 0.05$]. The mean relative reductions in LFC from baseline were 12.80% (placebo), 12.43% (ZSP1601 50 mg QD), 21.86% (ZSP1601 50 mg BID), and 25.50% (ZSP1601 100 mg BID). The reductions of LFC from baseline were dose-related. The maximal

**Table 1 | Baseline demographic, laboratory, and clinical characteristics**

| | | Pooled placebo (N = 9) | ZSP1601 | | |
|---|---|---|---|---|---|
| | | | 50 mg QD (N = 9) | 50 mg BID (N = 9) | 100 mg BID (N = 9) |
| Age, years | | 37.0 (32.0–40.0) | 34.0 (31.0–36.0) | 28.0 (25.0–38.0) | 30. 0 (24.00–38.0) |
| Sex | Male | 5 (55.6) | 9 (100) | 8 (88.9) | 7 (77.8) |
| | Female | 4 (44.4) | 0 (0) | 1 (11.1) | 2 (22.2) |
| Ethnicity, Asian (Chinese) | | 9 (100) | 9 (100) | 9 (100) | 9 (100) |
| Liver chemistries | ALT, U/L | 95.33 (21.12) | 90.89 (15.11) | 106.70 (22.94) | 132.34 (62.29) |
| | AST, U/L | 49.46 (12.55) | 41.19 (8.22) | 56.64 (16.69) | 68.46 (31.55) |
| | ALP, U/L | 79.49 (23.10) | 64.08 (22.05) | 103.97 (36.76) | 96.41 (20.51) |
| | GGT, U/L | 93.88 (44.08) | 66.19 (34.13) | 116.61 (84.35) | 84.76 (39.96) |
| LFC, % | | 24.39 (10.77) | 21.24 (7.75) | 21.00 (6.03) | 20.99 (9.05) |
| Noninvasive biomarkers | CAP, dB/m | 329.00 (28.31) | 337.33 (29.08) | 318.89 (44.56) | 321.63 (41.85) |
| | LSM, Kpa | 7.11 (1.66) | 6.51 (2.41) | 8.69 (6.63) | 6.53 (2.46) |
| | FIB4 | 0.72 (0.33) | 0.62 (0.36) | 0.76 (0.35) | 0.69 (0.30) |
| | APRI score | 0.49 (0.17) | 0.41 (0.14) | 0.59 (0.19) | 0.64 (0.33) |
| | FAST score | 0.52 (0.09) | 0.48 (0.16) | 0.51 (0.24) | 0.53 (0.17) |
| Lipids | TG, mmol/L | 2.92 (1.37) | 3.16 (2.08) | 3.95 (2.31) | 1.92 (0.83) |
| | TC, mmol/L | 5.25 (1.06) | 4.76 (1.07) | 5.83 (0.94) | 5.24 (0.89) |
| | LDL-C, mmol/L | 3.46 (0.78) | 2.90 (0.88) | 3.88 (0.90) | 3.43 (0.72) |
| | HDL-C, mmol/L | 1.13 (0.24) | 1.15 (0.30) | 1.15 (0.22) | 1.17 (0.23) |
| Blood glucose, mmol/L | | 6.22 (1.08) | 5.64 (1.52) | 5.64 (0.57) | 6.00 (0.95) |
| Blood insulin, µU/mL | | 128.23 (104.46) | 112.08 (64.13) | 139.08 (73.66) | 151.50 (83.30) |
| Inflammatory cytokines | TNF-α, pg/mL | 0.86 (0.26) | 0.74 (0.17) | 0.92 (0.30) | 0.64 (0.20) |
| | IL-6, pg/mL | 3.27 (1.82) | 1.64 (0.72) | 2.19 (2.40) | 2.37 (1.65) |
| | IP-10, pg/mL | 125.82 (61.30) | 51.81 (35.55) | 80.49 (52.20) | 115.77 (47.41) |
| | CK-18, pg/mL | 1008.56 (1283.75) | 509.70 (218.71) | 1509.87 (1390.87) | 810.53 (632.73) |
| Obesity parameters | Body weight, kg | 81.02 (8.94) | 83.92 (8.15) | 85.96 (13.19) | 81.10 (12.33) |
| | BMI, kg/m$^2$ | 29.47 (3.57) | 29.03 (2.76) | 30.32 (2.89) | 28.38 (3.04) |
| | Waist circumference, cm | 99.80 (6.96) | 102.03 (6.83) | 100.12 (5.77) | 97.02 (11.15) |
| | Abdominal circumference, cm | 102.23 (7.33) | 102.73 (6.94) | 101.22 (8.03) | 98.16 (11.76) |

Age was expressed as median (interquartile range); Categorical data (sex and ethnicity) were expressed as number (percentage), Other data were expressed as mean (standard deviation).
Inflammatory cytokines IL-22 and IL-10 were not detectable at baseline.

absolute and relative reduction in LFC from baseline was found in the ZSP1601 100 mg BID group. It was worth noting that ≥ 30% reduction in the LFC from baseline occurred in 4 patients (4/9) in the ZSP1601 100 mg BID group, 3 patients (3/9) in the ZSP1601 50 mg BID group, no patients in the ZSP1601 50 mg QD group, and 1 patient (1/9) in the placebo group. The results indicated that the proportion of patients with a ≥ 30% reduction in LFC from baseline was greater in the ZSP1601 50 mg BID and 100 mg BID groups versus the placebo group, although the difference was not statistically significant.

The results of secondary efficacy indicators were summarized in Supplementary Table 1. On the basis of assessment with FibroScan after the 28-day of treatment, CAP for liver steatosis only in the ZSP1601 100 mg BID group was significantly decreased, with the LS mean absolute change from baseline of −14.38 (95% CI, −43.80 to 15.03) dB/m (P = 0.0395) (Supplementary Table 1). LSM showed no significant change in all groups. Compared to the placebo group, the FAST score in the ZSP1601 100 mg BID group decreased significantly (P = 0.0379) (Supplementary Table 1). Changes in FIB4 measurement and APRI from baseline after completion of the 28-day treatment were not significantly different between placebo and ZSP1601 treatment groups. In terms of other secondary efficacy indicators, including lipid, blood glucose and insulin levels, inflammatory cytokines, and obesity parameter, there were no significant differences between the placebo and ZSP1601 treatment groups at the end of treatment, compared to baseline.

## Discussion

This randomized, double-blinded, placebo-controlled, multiple-dose phase Ib/IIa trial assessed the safety, pharmacokinetics, and efficacy of ZSP1601 in the treatment of patients with NAFLD. To the best of our knowledge, this is the first clinical trial to evaluate the efficacy of ZSP1601 in NAFLD. The results obtained in this exploratory small-scale study of short treatment duration are promising. Firstly, the efficacy of ZSP1601 in the treatment of NAFLD was assessed, and the major findings were as follows: (1) Serum ALT was significantly decreased from the baseline in ZSP1601 50 mg BID, and 100 mg BID versus placebo after the end of treatment; (2) The number and proportion of NAFLD patients with a reduction of ALT levels ≥ 30% from baseline was significantly higher in the ZSP1601 100 mg BID than the placebo as early as 22 days following the treatment; (3) The number and proportion of NAFLD patients with a ≥ 30% reduction in LFC from baseline was greater in ZSP1601 50 mg BID and 100 mg BID groups when compared with the placebo group; (4) Other exploratory efficacy indicators, such as FibroScan parameters, were decreased in response to ZSP1601 treatment. Secondly, ZSP1601 (50 mg QD, 50 mg BID, 100 mg BID) was safe and well-tolerated by the study subjects. Thirdly, pharmacokinetic profiles revealed the dose-proportional increase in exposure in ZSP1601 treatment groups.

At present, no pharmacological therapies are approved for NASH except that saroglitazar was approved for non-cirrhotic NASH by the Drug Controller General in India, and the treatment is restricted to

**Table 2 | Adverse events and adverse drug reactions to ZSP1601**

| Cohort | Placebo (N = 9) | ZSP1601 | | |
| --- | --- | --- | --- | --- |
| | | 50 mg QD (N = 9) | 50 mg BID (N = 9) | 100 mg BID (N = 9) |
| AE, n (%) | 6 (66.7) | 7 (77.8) | 6 (66.7) | 8 (88.9) |
| Grade ≥3 AEs | 3 (33.3) | 2 (22.2) | 0 (0) | 0 (0) |
| ADR, n (%) | 3 (33.3) | 4 (44.4) | 4 (44.4) | 7 (77.8) |
| Grade 2 ADRs | 0 (0) | 0 (0) | 0 (0) | 2 (22.2) |
| Elevated serum creatinine | 1 (11.1) | 0 (0) | 2 (22.2) | 2 (22.2) |
| T wave abnormality in ECG | 1 (11.1) | 0 (0) | 0 (0) | 0 (0) |
| ST-T segment changes in ECG | 0 (0) | 1 (11.1) | 0 (0) | 0 (0) |
| Diarrhea | 1 (11.1) | 2 (22.2) | 2 (22.2) | 1 (11.1) |
| Indigestion | 0 (0) | 0 (0) | 0 (0) | 3 (33.3) |
| Abdominal pain | 0 (0) | 2 (22.2) | 0 (0) | 0 (0) |
| Headache | 1 (11.1) | 0 (0) | 0 (0) | 4 (44.4) |
| Dizziness | 0 (0) | 1 (11.1) | 0 (0) | 0 (0) |
| Sinus arrhythmia | 1 (11.1) | 0 (0) | 0 (0) | 0 (0) |
| Sinus tachycardia | 0 (0) | 1 (11.1) | 0 (0) | 0 (0) |
| Limb pain | 0 (0) | 0 (0) | 0 (0) | 1 (11.1) |
| Myalgia or mus-cle pain | 0 (0) | 0 (0) | 1 (11.1) | 0 (0) |
| Anorexia | 0 (0) | 0 (0) | 0 (0) | 1 (11.1) |

*AE* adverse event, *ADR* adverse drug reaction, *N* number of subjects analyzed, *n* number of subjects who experienced at least one AE.

lifestyle modifications[10,13,22,23]. Globally, a wide range of potential drugs mainly targeting lipids and lipid-processing pathways, or inflammatory signaling pathways are current undergoing clinical trials, of which a small number of drug candidates have entered phase 3 clinical trial for NASH[13,32,35–38]. Indeed, this study of efficacy and safety of the first-in-class pan-PDE inhibitor ZSP1601 in patients with NAFLD has several strengths. Oral administration of ZSP1601 50 mg BID or 100 mg BID for 28 days showed significant reductions in serum ALT levels in the patients with NAFLD. Given that a reduction of ALT greater than 17 U/L has been shown to be associated with a histological response in patients with NASH[19,39–41], the alterations in serum ALT following a short duration of treatment with ZSP1601 50 mg BID or 100 mg BID could indicate reduced hepatocyte injury and may be clinically relevant to patients with NAFLD. It is worth noting that the beneficial effects of ZSP1601 on NAFLD were achieved after 28 days of continuous treatment, which is shorter than the longer treatment duration (usually 12 weeks or longer) utilized by other clinical trials. In addition, efficacy analysis demonstrated significant reductions in LFC as quantified by MRI-PDFF and some noninvasive markers for liver steatosis and fibrosis. We also assessed a range of secondary efficacy indicators after the 28-day treatment, including lipid levels, fasting glucose levels and insulin, inflammatory cytokines, and obesity parameters. The slight but not clinically meaningful changes were observed for some secondary efficacy indicators, and this could be explained, at least in part by the relatively short treatment period, which may not have been long enough to observe significant changes, especially for obesity parameters such as body weight, BMI, waist circumference, and abdominal circumference. As such, to observe significant and meaningful alterations in these secondary efficacy indicators to occur, a longer duration of treatment may be required. In terms of safety and tolerability, AEs related to ZSP1601 were transient mild to moderate, and there was no significant difference between ZSP1601 and placebo. These findings are in agreement with the results from the phase 1

**Table 3 | Plasma pharmacokinetic parameters of ZSP1601 and M3-5**

| Treatment | Day 1 | | | | Day 14 | | | | |
| --- | --- | --- | --- | --- | --- | --- | --- | --- | --- |
| | $t_{1/2}$ (h) | $T_{max}$ (h) | $C_{max}$ (μg/mL) | $AUC_{0-12h}$ (h μg/mL) | $T_{max}$ (h) | $C_{max}$ (μg/mL) | $AUC_{ss}$ (h μg/mL) | $R_{Cmax}$ | $R_{AUC}$ |
| **ZSP1601** | | | | | | | | | |
| 50 mg QD (n = 9) | 9.20 (2.56) | 2.50 (1.50–4.00) | 1.00 (0.208) | 7.59 (1.44) | 2.00 (0.750–3.00) | 1.32 (0.217) | 14.7 (3.48) | 1.33 (0.169) | 1.36 (0.0848) |
| 50 mg BID (n = 9) | – | 2.50 (0.250–4.00) | 1.11 (0.234) | 7.59 (0.948) | 2.00 (0.500–3.00) | 1.83 (0.330) | 14.7 (2.34) | 1.71 (0.457) | 1.95 (0.306) |
| 100 mg BID (n = 9) | – | 2.00 (0.750–3.00) | 2.33 (0.371) | 15.9 (1.56) | 1.00 (0.500–3.00) | 3.60 (0.469) | 27.8 (5.25) | 1.58 (0.327) | 1.75 (0.322) |
| **M3-5** | | | | | | | | | |
| 50 mg QD (n = 9) | – | 6.00 (6.00–12.0) | 0.266 (0.0415) | 2.43 (0.451) | 6.00 (2.50–8.00) | 0.403 (0.0599) | 7.05 (1.41) | 1.53 (0.180) | 1.57 (0.166) |
| 50 mg BID (n = 9) | – | 8.00 (6.00–11.5) | 0.232 (0.0430) | 2.14 (0.508) | 4.00 (2.00–8.00) | 0.630 (0.101) | 6.53 (1.09) | 2.77 (0.507) | 3.15 (0.618) |
| 100 mg BID (n = 9) | – | 6.00 (4.00–11.8) | 0.553 (0.114) | 5.24 (1.19) | 3.00 (1.00–6.00) | 1.33 (0.116) | 13.7 (1.55) | 2.50 (0.581) | 2.77 (0.874) |

Results were expressed as the arithmetic mean (standard deviation), except for $T_{max}$, which was presented as median (range). $t_{1/2}$, elimination half-life; $C_{max}$, maximum concentration in the plasma; $T_{max}$, the time of peak plasma concentration or the time needed to reach $C_{max}$; $AUC_{0-12h}$, the area under the plasma concentration–time curve from 0 to 12 h; $AUC_{ss}$, the area under the plasma concentration–time curve at steady state; $R_{Cmax}$, accumulation ratio $C_{max}$; $R_{AUC}$, accumulation ratio AUC.

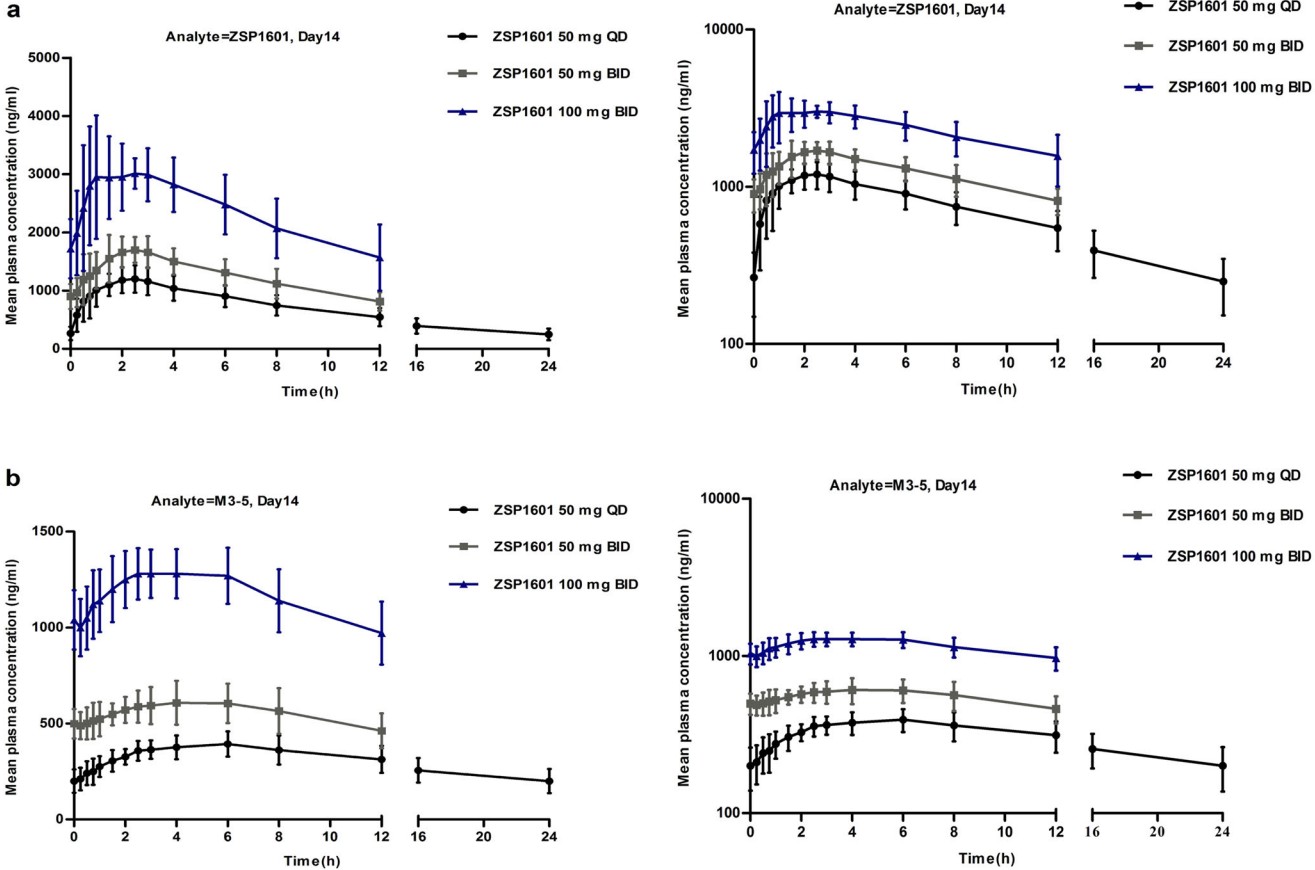

**Fig. 2 | Plasma concentrations of ZSP1601 and its metabolites in the three dose groups at different time points during the treatment period. a** Plasma ZSP1601 concentrations in the ZSP1601 50 mg QD (black line), 50 mg BID (gray line), and 100 mg BID (blue line) groups. Left panel, linear graph; Right panel, semi-log graph;

**b** Plasma concentrations of the metabolites M3–5 in the ZSP1601 50 mg QD (black line), 50 mg BID (gray line), and 100 mg BID (blue line) groups. Left panel, linear graph; Right panel, semi-log graph. Data are presented as mean ± standard deviation (SD). Source data are provided as a Source Data file.

clinical trial of ZSP1601 in healthy Chinese adults[27]. Collectively, the data regarding the use of ZSP1601 in the treatment of NAFLD obtained through this phase Ib/IIa randomized, double-blinded, placebo-controlled, multiple-dose trial showed significant reductions in the major noninvasive biomarkers that have been recognized to impact resolution of NAFLD and liver fibrosis. The findings in this study, together with those in the phase 1 clinical trial, demonstrate that ZSP1601 is a safe and promising new drug candidate for the treatment of NAFLD.

This study has some limitations. Firstly, the number of enrolled patients is relatively small, which limits the power of the study, thus it is likely that only very common and short-term adverse reactions would be detected. Also, we acknowledge that there was an imbalance in age and sex between the ZSP1601 and placebo groups, which may have resulted from the randomization process in this study with a small sample size. Although age and sex are not the main factor affecting efficacy, the analysis of results will take into account age and sex for a more comprehensive analysis in a larger Phase II/III clinical trial with a larger sample size. Secondly, liver biopsy is currently the gold standard for the definitive diagnosis of NASH and the management of NAFLD worldwide. Liver biopsies enable an accurate evaluation of liver steatosis, cell damage, inflammatory necrosis, and the degree of fibrosis, and histological findings based on liver biopsies have been widely used as the primary therapeutic endpoint in clinical trials of new drugs for NASH. In this study, the majority of the patients were diagnosed by clinical features with only two cases diagnosed by liver biopsy, and no comparison of histopathologic changes in study participants were made via liver biopsies before and after treatment. Thirdly, the treatment period of 28 days was relative short, thus we

were unable to determine potential long-term benefits of ZSP1601 in the attenuation of liver fibrosis in patients with NAFLD. Fourthly, the small number of women included in this study may be insufficient to detect even common adverse reactions in women. With the promising results gained from this phase Ib/IIa clinical trial, long-term treatment benefits of ZSP1601 should be further tested and sufficient toxicity data should be obtained in the future clinical trials using a larger sample size and extended treatment durations in patients with NASH.

In conclusion, the findings from this phase Ib/IIa randomized, double-blinded, placebo-controlled, multiple-dose trial in patients with NAFLD have suggested that ZSP1601 is well-tolerated and safe, and effective in the improvement of efficacy indicators for NAFLD, including statistically significant reductions compared with placebo in LFC, liver enzymes, and liver fibrosis. Thus, ZSP1601 is a potential novel medication that may be used in the treatment of NAFLD. Further clinical studies will be needed to assess long-term benefits and safety of ZSP1601 in a larger sample size of patients with NASH.

## Methods

This clinical trial has been registered at ClinicalTrials.gov, number NCT04140123.

This study and all amendments have been approved by the institutional review board or independent ethics committee of each study center [the First Hospital of Jilin University (Changchun, China), Nafang Hospital, Nanfang Medical University (Guangzhou, China) and Beijing Friendship Hospital, Capital Medical University (Beijing, China); approval numbers: 19Y110-007, NFEC-201909-Y4, 2019-PI-036-04]. All procedures were performed in accordance with the principles of the

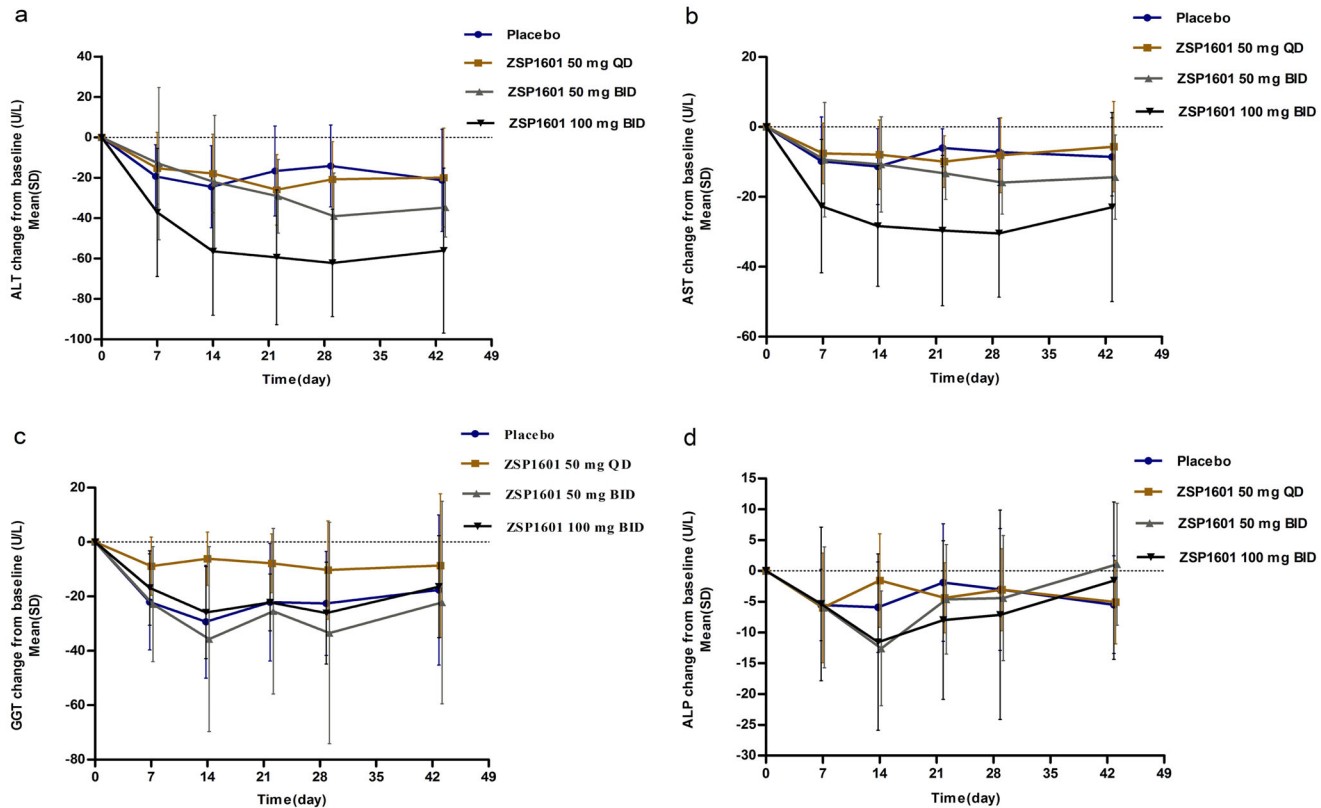

**Fig. 3 | Alterations in serum ALT, AST, GGT, and ALP levels following treatment with ZSP1601 in the NASH patients.** Time courses for changes from baseline in levels of ALT (**a**), AST (**b**), GGT (**c**), and ALP (**d**). Data are plotted as mean ± standard deviation (SD). Different groups are indicated in color (placebo, blue; 50 mg QD, brown; 50 mg BID, gray; 100 mg BID, black). Source data are provided as a Source Data file.

Declaration of Helsinki and Good Clinical Practice. Prior to the trial entry, all participants provided written informed consent.

## Patients and study design

In this phase Ib/IIa randomized, double-blinded, placebo-controlled, multiple-dose trial, a total of 95 patients with NAFLD were recruited and screened for eligibility in the First Hospital of Jilin University (Changchun, Jilin, China) and another two medical centers in China, including Nanfang Hospital, Nanfang Medical University (Guangzhou, Guangdong, China) and Beijing Friendship Hospital, Capital Medical University (Beijing, China) between June 2020 and June 2021, and the last patient completed follow-up in August 2021. Patients were diagnosed by liver biopsy with a NAFLD activity score (NAS) of 4 or more, or by clinical features of NAFLD that fulfilled the following two criteria: (1) Alanine aminotransferase (ALT) ≥ 1.5×ULN (male 75 U/L, female 60 U/L) for two examinations with an interval at least 7 days; (2) Body-mass index (BMI) ≥ 25 kg/m². Eligible NAFLD patients were 18–65 years of age, with fatty infiltration of the liver based on abdominal ultrasound findings, and baseline hepatic fat fraction of at least 10% by MRI-PDFF. The schematic diagram of patient enrollment is illustrated in Fig. 1. Before the clinical trial-related procedures were performed, patients were assessed for eligibility and enrolled by participating investigators at the three trial centers. During enrollment, patients with any of the following conditions were excluded from this clinical trial: (1) history or presence of chronic liver diseases other than NAFLD; (2) cirrhosis, hepatic decompensation, or hepatocarcinoma; (3) presence of other chronic medical conditions, including type 1 diabetes mellitus, uncontrolled T2DM, hemoglobin A1C (HbA1c) ≥ 8.0%, cardiovascular disease; (4) taking medications that may have confounding effects on the trial results, such as thiazolidinediones and *Silybum marianum*; (5) excessive alcohol consumption (more than 20 or 30 g of alcohol every day for men or women, respectively) for three consecutive months within one year before screening. There were no enrollment restrictions by center and participants in each dose group were distributed across the three participating centers.

## Patient allocation and interventions

The PLAN statistical procedure in SAS9.4 software was used to randomly allocate the patients to interventions and interventions to blocks in a 3:1 ratio for each dose group (nine patients to receive ZSP1601 and three patients to receive placebo in block randomization). The randomization sequence was generated by an independent statistician. A sheet with the information regarding the allocation and identification numbers was placed in an envelope, which was then sealed. During the study, investigators, participants, sponsors and any combination of them are unaware of allocation of the treatment drug.

ZSP1601 tablets (two dosage forms: 25 mg and 100 mg) were obtained from Guangdong Raynovent Biotech Co., Ltd (Guangzhou, Guangdong, China). This clinical trial consisted of three dose cohorts with multiple doses of ZSP1601, including 50 mg once daily (QD), 50 mg twice daily (BID) (time interval of 12 ± 2 h), and 100 mg BID (time interval of 12 ± 2 h). Each cohort included twelve patients, of which nine patients were treated with ZSP1601 50 mg QD, 50 mg BID, or 100 mg BID, and three patients received matching placebo orally once or twice daily for 28 days. Patient compliance was monitored in this study. Briefly, at the research center, two researchers were responsible for the study medication administration at a fixed time. Before discharge from the research center, patients were instructed to take the study medication at home and record the medication status. At each time when the study subject returned to the research center for a visit,

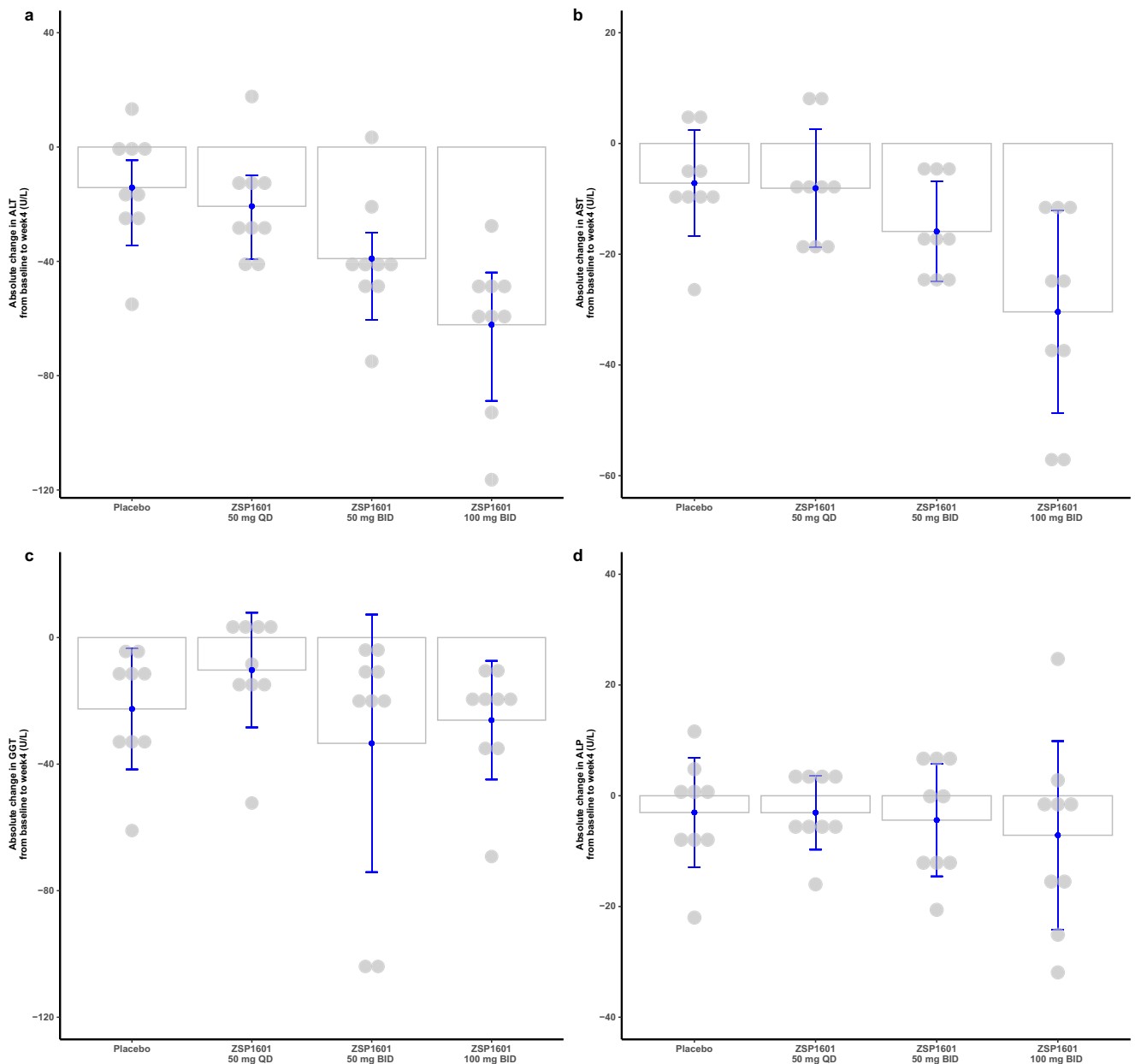

**Fig. 4 | Alterations in serum ALT, AST, GGT, and ALP levels from baseline upon completion of the 28-day treatment in the NASH patients.** Bar graphs represented means with blue error bars reflecting standard deviation (SD) with absolute changes in ALT (**a**), AST (**b**), GGT (**c**), and ALP (**d**) levels from baseline as the dependent variable. Each gray dot represents an individual (each group, $N = 9$). Source data are provided as a Source Data file.

the researchers verified the medication status. Immediately upon completion of the 28-day treatment, patients received treatment-free follow-up for 14 days.

## Laboratory and clinical examinations

Blood samples were collected from study subjects at baseline, during treatment (day (D)7, D14, D22), upon completion of the 28-day treatment (D29), and 14 days after treatment with ZSP1601 (D43). Liver biochemical tests were conducted to examine serum ALT, aspartate aminotransferase (AST), gamma-glutamyl transferase (GGT), and alkaline phosphatase (ALP). Serum biomarkers included cytokeratin-18 (CK-18) as a biomarker of cell death and mitochondrial dysfunction, inflammatory cytokines including TNF-α, interleukin (IL)−6, IL-10, and IL-22, and interferon-gamma inducible protein 10 (IP-10), a chemokine important in the modulation of innate and adaptive immune responses.

## Safety assessment

Safety and tolerability assessments of ZSP1601 were undertaken for each patient throughout the study. Adverse events (AEs) need to be collected during the study. The Common Terminology Criteria for Adverse Events (CTCAE 5.0) was used to grade the severity of the AEs, according to five categories of grades: Grade 1 (mild), Grade 2 (moderate), Grade 3 (severe), Grade 4 (life threatening), and Grade 5 (death). The AEs identified as severe were reported using the Serious Adverse Event Report Form. The attribution of all AEs was determined by the investigators. Based on CTCAE 5.0, dose escalation will be halted if any of the following conditions are met: (1) More than half of the subjects experience grade 2 or higher drug-related adverse events, (2) More than a quarter of the subjects experience grade 3–4 drug-related adverse events, or (3) a drug-related serious adverse event (SAE) occurs. The next highest dose level was initiated after a 14-day safety observation of the previous by principal investigator and sponsor.

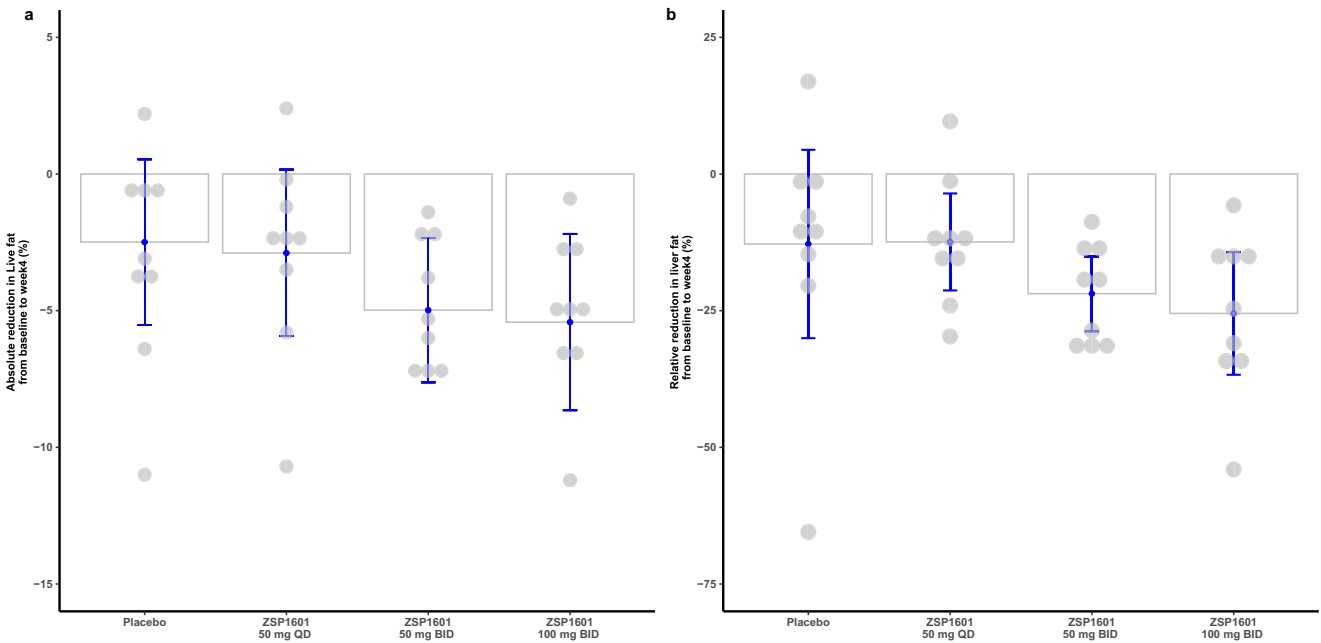

**Fig. 5 | Alterations in liver fat content from baseline assessed by MRI-PDFF upon completion of the 28-day treatment. a** Columns representing LS means with blue error bars reflecting two-sided 95% CIs from an ANCOVA model, with absolute change from baseline as the dependent variable and the baseline values as a covariate; **b** Relative changes from baseline, **c**olumns representing means with blue error bars reflecting standard deviation (SD). Each gray dot represents an individual (each group, $N = 9$). Source data are provided as a Source Data file.

## Phamacokinetics

Blood samples for pharmacokinetic (PK) assessments were collected on days 1, 14, and pre-dose on days 2, 3, 4, 5, 6, 7 and 12. The main PK parameters of ZSP1601 at day 1 were the elimination half-life of ZSP1601 ($T_{1/2}$), maximum concentration in the plasma ($C_{max}$), the time of peak plasma concentration or the time needed to reach $C_{max}$ ($T_{max}$), and the area under the plasma concentration-time curve from 0 to 12 h ($AUC_{0–12}$). At day 14, parameters measured were $C_{max}$, $T_{max}$, the area under the plasma concentration-time curve at steady state ($AUC_{ss}$), and the accumulation ratio $C_{max}$ ($R_{Cmax}$). At day 14, the main PK parameters of metabolites M3–5, including $C_{max}$, $T_{max}$, $AUC_{0–12}$ at day 1, and $C_{max}$, $T_{max}$, $AUC_{ss}$, and $R_{Cmax}$ were also determined. Renal clearance ($CL_R$) of ZSP1601 and its metabolites M3–5 were also calculated in the study subjects.

## Efficacy evaluation

Efficacy evaluations were performed in subjects who completed the trial. The exploratory efficacy outcomes were changes from baseline in liver biochemical tests (ALT, AST, ALP, and GGT), LFC as examined by MRI-PDFF, FibroScan values with two parameters (LSM and CAP using transient elastography) for evaluation of liver fibrosis and steatosis, FAST score, AST to platelet ratio index (APRI), Fibrosis 4 (Fib 4), inflammatory cytokines (TNF-α, IL-6, IL-10, and IL-22) and other potential noninvasive serum biomarkers (IP-10, CK-18), lipid panel tests [total cholesterol (TC), triglyceride (TG), high-density lipoprotein cholesterol (HDL-C), low-density lipoprotein cholesterol (LDL-C)], serum insulin, glucose, and calculate Homeostatic Model Assessment of Insulin Resistance (HOMA-IR). Obesity parameters [body weight, body-mass index (BMI), waist circumference, abdominal circumference]. All the MRI-PDFF and LSM were measured in the study subjects at baseline and after the end of the treatment at day 29 (D29) in the placebo, ZSP1601 50 mg QD, ZSP1601 50 mg BID, and ZSP1601 100 mg QD groups. Liver biochemical tests (serum ALT, AST, ALP, and GGT), and other laboratory tests were examined at baseline, during treatment at day 7, day 14, day 22, upon completion of the 28-day treatment (D29), and 14 days after treatment with ZSP1601 (D43) in the

placebo, ZSP1601 50 mg QD, ZSP1601 50 mg BID, and ZSP1601 100 mg BID groups.

## Statistical analysis

Statistical analysis was conducted using SAS statistical software for Windows, Release V.9.4 (Armonk, NY, USA). Phoenix WinNonlin version 7.0 (Certara, Princeton, NJ, USA) was used for statistical analysis of the pharmacokinetic parameters. Descriptive statistics were presented using the mean and standard deviation, median, and interquartile range. An Analysis of Covariance (ANCOVA) model was used to calculate the least square mean (LSmean), the LSmean difference and the 95% confidence interval (95% CI) of efficacy outcomes between groups and time points with Tukey's multiplicity correction. A Model check was performed to assess on statistically significant indicators, and the Kruskal–Walis test was used for indicators that did not meet the assumptions of normality and homogeneity of variance to further confirm the statistical differences. The Wilson score method was used to calculate 95% CI and the Newcombe–Wilson Score method was used for calculating confidence intervals of the difference in the proportion of patients achieving ≥ 30% reduction in liver biochemical parameters and LFC from baseline between two groups. All the statistical tests were two-sided, and a 5% type I error was used to reject null hypotheses.

## Reporting summary

Further information on research design is available in the Nature Portfolio Reporting Summary linked to this article.

## Data availability

The data that support the findings of this study are available on request from the corresponding author. Individual deidentified participant data (including data dictionaries) will be shared. Individual participant data that underlie the results reported in this article, after deidentification. Study protocol will be available as supplementary file. The data will become available at beginning 3 months and ending 3 years following article publication. Proposals should be directed to

dingyanh@jlu.edu.cn. All proposals for data will be reviewed and will ensure that the proposals are complete and valid, and that the data are available, consistent with participant privacy and informed consent by the corresponding author, and responses will be provided within two months. To gain access, data requestors who provide a methodologically sound proposal will need to sign a data access agreement. Source data have been provided with this paper and its supplementary information files.

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

## Acknowledgements

This work was financially supported by the capital construction funds within the provincial budget in 2020 (innovation capacity construction, project: 2020C038-1) and China National Major Scientific and Technological Special Project (Project No: 2018ZX09201002-002-001). The funders had no role in the study design, data collection, data analysis, decision to publish, and manuscript preparation. We would like to thank all patients for their willingness to participate in this clinical study and Investigators.

## Author contributions

Y.H., H.L., X.Z., Y.D. and J.N. were involved in the conception, design, and analyses of the data; Y.H., H.L., H.Z., X.C., J.C., Z.X., H.Y., R.D., Y.P., J.L., X.L., D.W., L.Z., D.C., H.J., D.Q., A.Y., J.Lou, X.Z., Y.D. and J.N. performed the clinical trials, acquisition and interpretation of the data; Y.H., Y.D., and J.N. participated in the drafting of the paper; Y.H., H.L., H.Z., X.C., J.C., Z.X., H.Y., R.D., Y.P., J.L., X.L., D.W., L.Z., D.C., H.J., D.Q., A.Y., J.Lou, X.Z., Y.D. and J.N. contributed to revising the paper critically for intellectual content and approving the final version to be published; Y.H., H.L., H.Z., X.C., J.C., Z.X., H.Y., R.D., Y.P., J.L., X.L., D.W., L.Z., D.C., H.J., D.Q., A.Y., J.Lou, X.Z., Y.D. and J.N. agree to be accountable for all aspects of the work.

## Competing interests

H.L., X.C., Y.P. and J.L. are employed by Guangdong Raynovent Biotech Co., Ltd. All other authors declare no conflict of interest.
