## [Peer Review File · Nature Communications]

ZSP1601, a novel pan-phosphodiesterase inhibitor for the treatment of NAFLD, A randomized, placebo-controlled phase Ib/IIa trialDear Editor,

As a statistician, I mainly focused on the quantitative aspects of the manuscript.

Comments/questions:

1/ 'Multi-dose escalation' & 'ascending doses':

These concepts, used on lines 127, 155, 232, 475, 427, 449 and 551, usually refer to adaptive trials in which allocation to the dose of the c th cohort depends on data collected on cohorts 1 to $c-1$ (like seemingly considered in the phase I trial mentioned on line 115). It doesn't seem to be the case this phase Ib/IIa. Please clarify or prefer 'multiple-dose trial' without mentioning 'escalation' or 'ascending'.

2/ 'Multiplicity':

Between lines 167 and 222, more than 50 outcomes are described. A statement regarding multiplicity correction for primary and secondary outcomes in the 'statistical analysis' section would be welcomed.

For example, in Table 4, the mean of each treatment group is compared to the reference one for ~ 15 outcomes, *possibly* (unclear) with a multiplicity correction at the outcome level. This still leaves a rather high chance of detecting differences when there is none (like for CAP and FAST, for example). Therefore, a factual statement warning the reader of the 'large' chance of false positives in secondary outcomes would be useful for the reader to take the statements of lines 362-363 (outcome CAP) and 365-366 (outcome FAST) with caution.

3/ 'Sample size':

Phase Ib/IIa trials typically consider small sample sizes. Could the authors still mention why $n=36$ (Figure 1) was preferred to $n=24$ or $n=45$ for example, or, if they started with $N=95$ (Figure 1), why 95 patients were initially considered and not 80 or 120. Does it correspond to a number of patients per unit of time in the centres of interest?

4/ 'Statistical analysis':

Some (significant) rewriting of the statistical analysis may be of benefit to the reader.

4a) MEANS:

4a-1/ lines 229 to 233: "LS means": Authors seems to be performing "tests of equality of means between groups and/or time points with Tukey's multiplicity correction" (this sounds much clearer 'LS means corrected by Tukey's method') depending on the analysis. From their description, it is unclear what analysis was actually performed. For example

* in Table 4: are the 'LSmean (95%CI)' and the LSmean difference (95%CI) corrected for multiplicity?

* in Figures 2 and 3: how are the confidence intervals defined and are there corrected for multiplicity?

Therefore, it may be useful to list all ways of getting estimates and confidence intervals (with and without confidence intervals) and to specify in which Tables and Figures they were used.

4a-2/ why Tukey?

Also, the authors seem to be using Tukey's HSD (https://en.wikipedia.org/wiki/Tukey%27s_range_test) to correct for multiplicity (please confirm). This method controls for the familywise error rate when considering comparisons between **all pairs** of groups (ie, 10 comparisons in total for 4 groups). However, the authors only seem to focus on the comparison of each treatment arm with the reference one (ie, 3 comparisons). In such a case, a Dunnett like multiplicity correction (https://en.wikipedia.org/wiki/Dunnett%27s_test) or multiplicity correction for specific contrasts of parametric models are likely to be more powerful.

4a-3/ model check

In small sample, the coverage of 95% confidence intervals based on the linear model **strongly** depends on the respect of linear model assumption (normality, homoscedasticity). A statement claiming that model checks were performed for each outcome and showed respect of the normality and homoscedasticity assumptions on the chosen scale (original or log scale, for example) would be useful.

4b) PROPORTIONS:

4b-1/ lines 233-234: "the Wilson method was used to calculate 95% CI": Authors seems to use Wilson score intervals to define confidence intervals for binomial proportions (<https://doi.org/10.1214/aos/1015362189> formula (3.1)). Please confirm, specify if a continuity correction was used, mention that this is for proportions, and provide a reference.

4b-2/ lines 234-236: "Newcombe-Wilson score method": This seems to refer to [https://doi.org/10.1002/\(SICI\)1097-0258\(19980430\)17:8<873::AID-SIM779>3.0.CO;2-I](https://doi.org/10.1002/(SICI)1097-0258(19980430)17:8<873::AID-SIM779>3.0.CO;2-I) . Please confirm, specify if a continuity correction was used and provide a reference. again, please add a comment regarding multiplicity so that it is clear to the reader when this was used.

4c) OTHER:

lines 237-238: The authors probably mean that the 'statistical tests' were two sided (not the p-values) and that a 5% type I error was used to reject null hypotheses.

5/ 'Demographics' and 'Randomisation':

Authors claim, on line 251 to 253, that demographic characteristics of the study subjects at baseline were comparable between treatment and control. Some may dispute this by pointing out that the proportion of females is (significantly, according to a Fisher test) larger in the control group (4/9) than in the treatment arms (3/27). Some may also claim that, given the statistics provided in Table 1, the patients of the control group are often older than patients in the treatment arms. It may be useful to explain how age and sex were randomised, to possibly consider a sensitivity analyses controlling for such imbalances (we recognise this may be difficult given the small sample size) and to display age (y-axis) per group (x-axis) in a plot (1 point per participant, coloured coded by sex) as supplemental material.

If authors believe that such unbalances are not problematic as sex and age are not believed to be confounders, they should say so.

6/ 'Target population':

All patients are Chinese. It may be useful to clarify if the target population is 'Asian' or if similar effects are expected on other ethnic groups (sorry if this was mentioned and if I failed to see it).

7/ 'Efficacy' section:

7a) consider splitting your efficacy section (lines 302 to 372) between primary and secondary outcomes (as defined in the protocol) by using sub sections to clearly highlight what your primary conclusions are.

Indeed, the fact that you mention ALT, AST, LFC on line 303 but display ALT, AST GGT and ALP in Figure 3 but only focus on ALT and AST in Figure 4 is *confusing*. It would be helpful to display the same information for all primary outcomes (both in the different Tables and Figures).

7b) lines 310: could you specify where this number come from (Table, Figure)

7c) lines 312 to 324: if these analyses are exploratory (inspired by previous significant results and trying to explain them more deeply by 'following the data'), please say so. It is hard to track the number of tests you performed.

7d) lines 326 to 360: same as for ALT.

8/ 'Discussion' section:

Given your large familywise error rate, probably try to remain on the cautious side regarding your conclusions, especially when commenting on secondary outcomes.

9/ 'Table 1':

The legend notes that 'Data are expressed as n(%), mean(SD) or median(IQR)': It is unclear when mean(SD) or median(IQR) were used. It may be useful to indicate this more precisely.

10/ 'Tables 2 and 3':

- * it may be useful to specify what is shown (mean(SD) or n(%) or median(IQR)) and where.
- * in Table 3, the C max value of the first treatment group at day 1 seem to exactly equal 1. Were data standardised? If yes how?

11/ 'Table 4'

A legend in which you indicate why some results are presented with bold fonts would be useful.

12/ 'Figure 2':

very naively, I don't seem to spot the difference between the left and right panels in Figure 2A and 2B (same titles, same x-axes, same y-axes). If this relates to a replication study, maybe consider saying so in the legend, reflecting this in the title (experiment 1, experiment 2), and using the same y-axes in both panels to make comparison easier.

13/ 'Figure 3':

- * in the legend, you forgot to mention you also display GGT (C) and ALP (D).
- * the confidence intervals overlap: maybe consider shifting them slightly on the x-axis to allow comparisons.

14/ 'Figure 4':

- * your sample size per group is small enough to show all the individual data points. Please do so.
- * remember to indicate in the stats section if multiplicity correction per outcome in this analysis. If so, probably have this reflected on lines 308 and 309 by preferring adj. p-value to p-value.

15/ 'Data sharing':

Good to hear authors intend to share data. Could they maybe specify how and when?

16/ Editing:

16a) Lines 40, 156, 158: the abbreviations QD and BID are used without explanation. On line 156 and 158, the third treatment seems to be different: '100 mg BID' on line 156, '100 mg QD' on line 158.

16b) Lines 39 to 41: It may be clearer to the reader to specify more clearly the different cohorts. For example "[...], were enrolled in three cohorts: (i) 50 mg once daily, (ii) 50 mg twice daily and (iii) 100 mg once daily. In each cohort, patients were randomly assigned to the drug or matching placebo with a 3:1 ratio. The primary efficacy outcome [...]". If using (i), (ii) and (iii) as suggested above on lines 39 to 41, probably do the same on lines 156 and 158.

16c) Lines 45, 131 and : 'ALT' and 'AST' are defined later on page 10 only. It would be useful to define them when first using these abbreviations

16d) Line 113: Probably start a new paragraph with the word 'recently'

16e) Line 303: you seem to also consider GGT and ALP in Figure 3. Maybe add it here

16f) Line 617-620: left and right panels instead of upper and lower ones.

REVIEWER COMMENTS

Reviewer #2 (Remarks to the Author):

This is a phase Ib/IIa randomised trial of a pan-phosphodiesterase inhibitor, ZSP1601, in patients with nonalcoholic steatohepatitis. The study sought to evaluate the tolerability, safety, pharmacokinetics and early pharmacodynamics of the drug using a dose-escalation design, and also provides preliminary efficacy data with regards to liver biochemistry biomarker and imaging outcomes.

Major comments

1. The study was prospectively registered with primary outcomes relating to safety and tolerability. However, the reporting focuses on efficacy. Although the efficacy data warrants reporting, it should not be the main focus of the article and should be treated with appropriate caution given the small number of participants. The full article needs to be revised with this in mind, but in particular: the title should mention safety and tolerability; the summary should report these findings and in greater detail before mentioning the efficacy findings; and the primary and secondary outcomes should be explicitly identified in the methods. The mention of “Primary efficacy outcomes” in the summary (Line 41) is somewhat misleading – some of these outcomes were registered as secondary outcomes, and some were not registered at all (although they are mentioned in the protocol); in the main text, they are described as exploratory. I suggest that the statement of the aims at the end of the introduction should be clearer about the main aim vs secondary aims, too.
2. The discussion of subgroups needs to be treated with greater caution, given the very small numbers and the fact that non subgroup analyses were prespecified (as far as I can see, in the registration). When discussing subgroups, please give n, effect estimates, and CIs, not just p-values
3. Please provide details of the randomisation process, including who generated the allocation sequence and the method used; the method used to implement allocation (to ensure concealment); who enrolled participants and who assigned the interventions. Please also indicate any restriction by centre – whether participants in each dose-group were

distributed across the centres, or whether each of the three centres dealt with one of the three dose-groups only.

4. Methods, Safety assessment: Please describe the process for attributing adverse drug reactions.

5. Figures 4 and 5: these data would be better to displayed in dot plots to avoid information loss (see DOI: [10.1371/journal.pbio.1002128](https://doi.org/10.1371/journal.pbio.1002128) and DOI: [10.1161/CIRCULATIONAHA.118.037777](https://doi.org/10.1161/CIRCULATIONAHA.118.037777))

6. Please discuss the limitations of the study with regard to the primary safety outcome – the small number of participants and short duration means it is likely that only very common and short-term adverse reactions would be detected. Please also comment on the small number of women included – this may be insufficient to detect even common adverse reactions in women.

7. The declaration of interests states: “All authors declare no conflict of interest.” However, four of the authors list their affiliation as the company that developed the drug. Please review, and be sure to declare all relevant interests that could represent a conflict (actual or perceived).

8. In some places, the language, although grammatically correct, is either ambiguous in meaning or reports something other than what I think the authors are trying to say, specifically:

- Lines 38-41: The description of the trial arms in the summary is unclear – I would think (and the protocol seems to indicate) that group 1 would comprise 9 participants on 50 mg QD and 3 on placebo; then the second arm would introduce the next dose (9 participants on 50 mg BID, 3 on placebo) started after a time-lag, etc – but the summary implies each cohort of 12 had 3 participants on each dose + 3 on placebo, and methods section is ambiguous. Please clarify this. It might be helpful to mention the criteria for instigating the next dose-arm. A figure similar to the dose escalation flowchart from the protocol would be helpful, either as a figure in the main text, or a supplementary file.

- Line 44: meaning of “trend” is unclear here. If you’re trying to say that there were non-significant differences, it would be better to talk about observed difference but acknowledge uncertainty in the estimates and that it could be consistent with random variation.
- Line 51: “Exposure to ZSP1601 and its metabolites showed dose-proportional increases.” – is this still talking about adverse events, or pharmacokinetics? (increases in what?)
- Line 166: “Liver biochemical tests included...” – “included” implies that there were other liver biochemical tests conducted that are not listed. Please list all tests (and similarly for biomarkers).
- Lines 318-322: “When stratified by BMI, a significant reduction of serum ALT levels was observed in both ZSP1601 50 mg BID and 100 mg BID groups when compared to the placebo group (P=0.0002 and P=0.0011) in patients with a BMI \geq 28 kg/m².” – “a significant reduction” implies a null hypothesis of no reduction between baseline and post-drug. “compared to the placebo group” implies the null hypothesis is no difference in the size of the reduction between experimental and placebo drugs. Please clarify which is being assessed – either, “there was a significant reduction in [variable] in groups x and y but not in group z”, or “there was a significantly larger reduction in [variable] in both group x and group y compared with in group z”. I think it’s the latter (which is the more appropriate comparison, by my understanding).
- Similarly, Line 329- “Compared with patients who received placebo, AST level were significantly decreased from baseline...” I think you mean, “compared with in patients who received placebo, AST levels showed a significantly greater decrease from baseline in patients treated with...[active drug]”
- Line 468-474: The data sharing statement is unclear. “All data will be reviewed...” – do you mean that all requests for data access will be reviewed and evaluated as described? Please indicate to whom requests for access should be directed.

Minor comments

1. The “reporting summary” indicates that a completed CONSORT checklist should be provided with the submission, but I could not find this in the materials provided. Please supply a CONSORT checklist, ensuring all items are fully reported.

2. Please mention in the title that this is a randomized, placebo-controlled phase Ib/IIa trial, for ease of searching and data extraction.

3. Lines 112-113: “In preclinical studies, ZSP1601 demonstrated greater therapeutic effectiveness in comparison with PTX” – needs a citation (also, what is “therapeutic” meaning here, given that this is referring to a preclinical study?)

4. Lines 119-121: In establishing the need for your study, you state that: “further clinical trials to evaluate the efficacy of ZSP1601 in the treatment of NASH” are warranted – but the primary outcome of this study was safety, not efficacy. I’d suggest mentioning/justifying needing to evaluate safety and PK in patients with NASH (as opposed to healthy volunteers).

5. Line 133-134: “fatty liver as confirmed by abdominal ultrasound” – what were the criteria for this?

6. Line 141: “et al.” is vague – please specify all exclusion criteria

7. Line 147: please provide the ethics approval number(s)

8. Interventions: Please indicate the time-lag of ascending doses and the criteria for instigating (or not instigating) the next dose-group

9. Interventions: Please indicate whether and how compliance was monitored

10. Line 189: Please cite or describe the tool used for documentation

11. Line 194: “as reported previously” – please provide a citation.

12. Line 201: “including” – please list all

13. Line 215: “etc” – please list all

14. Please justify dichotomising ALT, and indicate the basis for choosing $\geq 30\%$ reduction from baseline as an outcome
15. Table 2: If n is the number of events, as indicated in the footnote, then the numbers do not add up. Possibly it is the number of participants experiencing one or more event?
16. Figures 3 and 5: please indicate in the figure legend what the error bars represent
17. Line 315: "There was an obviously dose-dependent tendency" – maybe "apparent" or "possible" dose-dependence would be better, given the small numbers and lack of statistical significance (and low power). (And "tendency" should be removed).
18. Line 332-335: "The proportion of patients with reduced AST levels $\geq 30\%$ from baseline was greater in the ZSP1601 100 mg BID group compared with the placebo group, whereas there was no significant difference between ZSP1601 treatment groups and placebo groups." – do you mean no significant difference between the 50 mg QD or the 50 mg BID group and the pooled placebo group? Also, please don't just say "not significant", please give the numbers and p-value.
19. Paragraph beginning line 337, "Alterations in other liver chemistries..." – too much focus on significant/non-significant dichotomy – better to discuss the scale of the difference and precision of estimate.
20. Lines 345-347: "[LS mean (95% CI), -2.49(-5.52 to 0.54), -2.89(-5.94 to 0.16), -4.98(-7.62 to -2.33) and 347 -5.42(-8.64 to -2.19) in placebo and ZSP1601 groups, respectively" – please specify which ZSP1601 group each value refers to.
21. Lines 391-392: "there was no significant difference in safety profiles between ZSP1601 and placebo groups" – unless I'm missing something, you didn't apply any statistical tests here, so please change the wording to avoid the term "significant".
22. Lines 433-435: "Firstly, the number of enrolled patients is relatively small, which may

lead to less statistical significance between the ZSP1601 and the placebo groups.” – This is a rather upside-down way of looking at it, and suggests the only concern is hitting “ $p < 0.05$ ” rather than evaluating the strength of the evidence for a clinically meaningful effect. The small sample size limits the power of the study to detect differences, and means the efficacy results need to be interpreted with caution.

23. Please indicate the role of the funders (or lack thereof) in the study design, data collection, analysis, decision to publish, and manuscript preparation.

Reviewer #3 (Remarks to the Author):

This manuscript focus on the ZSP1601 therapy for NASH patients. The manuscript is easier to follow, the design of the experiments is adequate and the experiments are executed correctly, but it requires the following corrections and explanations:

Point 1. Line 153, ZSP1601 tablets are 25mg and 100mg, or are they 50mg and 100mg?

Point 2. Line 165 correct typo.

Point 3. Clarify why 14 days after treatment is a good time point.

Point 4. The authors do not mention in Safety and tolerability results the $AE \geq 3$ (severe).

Point 5. In discussion section, the authors indicate statistically significant reductions in markers of inflammation; explain this point and the results that prove this conclusion.

Point 6. Explain the CAP meaning.

Point 7. Line 624 AST, aspartate aminotransferase.

Point8. Figure 3. C and D are missing in the figure legend.

Point9. Figure 4. Correct the letters of the figures, and explain what ANCOVA is in the statistics methods.

Point10. Figure 2. Indicate the figures differences (linear and semi log in the figures), they seem the same graph but with the Y-axis changed.

Reviewer #4 (Remarks to the Author):

The authors performed a 28 day study of ZSP1601 in 36 subjects (3 cohorts of 12 subjects each at 3 different escalating doses of drug) with NAFLD. Safety, tolerability and key non-invasive tests to include liver chemistry tests and liver fat content as measured by both CAP and MRI-PDFF were assessed. The authors demonstrated in this short trial with a small number of subjects that the drug was generally well tolerated and there was a dose response to ALT, AST and liver fat content.

This is a very small, proof of concept study with a novel mechanism in NAFLD subjects. The main concern with the data is the small # of subjects (only 3 pbo subjects and 9 treated subjects in each cohort). Additionally, treatment was limited to 28 days.

I have several additional comments relative to this study

1. The authors should narrow the focus of the background to their specific mechanism of action and minimize discussion of other mechanisms such as FXR, THR-beta, vitamin E, etc. In addition, there should be an explanation of what is expected to improve by working through this mechanism. While I understand you are looking at ALT as a marker of inflammation, why MRI-PDFF? What rationale is there to assess liver fat content with your mechanism? Do you impact fibrosis? This should be mentioned as well.
2. The title is not accurate. You should replace "NASH" in the title with "NAFLD" as this is not a biopsy proven NASH cohort. In fact, very little is done non-invasively to establish the diagnosis of NASH outside of elevated ALT, BMI \geq 25 kg/m² and liver fat by imaging.
3. Define excessive alcohol use
4. What defines a "normal" ALT?
5. Figure one is your consort diagram. It would be nice to also include a study design figure.
6. While this is a 28 day study, did any patients lose weight and if so, is this accounted for in

the non-invasive test assessments?

7. Please do not use "liver cirrhosis" together. This is an oxymoron

8. Please have your paper reviewed for English grammar

9. The discussion should include commentary of next steps, including next phase studies and is there sufficient tox data to support the trial duration.

Point-by-Point Responses: NCOMMS-22-49299A

Reply to Review #1:

Dear Editor,

As a statistician, I mainly focused on the quantitative aspects of the manuscript.

Comments/questions:

1/ 'Multi-dose escalation' & 'ascending doses':

These concepts, used on lines 127, 155, 232, 475, 427, 449 and 551, usually refer to adaptive trials in which allocation to the dose of the cth cohort depends on data collected on cohorts 1 to c-1 (like seemingly considered in the phase I trial mentioned on line 115). It doesn't seem to be the case this phase Ib/IIa. Please clarify or prefer 'multiple-dose trial' without mentioning 'escalation' or 'ascending'.

Response: We are grateful to you for providing us with thoughtful comments and suggestions, which have helped us in improving our manuscript. With respect to the issues on the quantitative aspects, we have made the following amendments or explanations.

Actually, this is an ascending dose study and we have made it clearer in Figure 1. In addition, we have checked the entire manuscript and made revisions as well.

2/ 'Multiplicity':

Between lines 167 and 222, more than 50 outcomes are described. A statement regarding multiplicity correction for primary and secondary outcomes in the 'statistical analysis' section would be welcomed.

For example, in Table 4, the mean of each treatment group is compared to the reference one for ~15 outcomes, *possibly* (unclear) with a multiplicity correction at the outcome level. This still leaves a rather high chance of detecting differences when there is none (like for CAP and FAST, for example). Therefore, a factual statement warning the reader of the 'large' chance of false positives in secondary outcomes would be useful for the reader to take the statements of lines 362-363 (outcome CAP) and 365-366 (outcome FAST) with caution.

Response: Thank you for your comment. We performed multiplicity correction for all the efficacy indicators (secondary outcomes) were presented in figures, tables and source data. In light of your comment and suggestion, we have included the additional information about the multiplicity correction under the heading of Statistical analysis in Methods section.

3/ 'Sample size':

Phase Ib/IIa trials typically consider small sample sizes. Could the authors still mention why n=36 (Figure 1) was preferred to n=24 or n=45 for example, or, if they started with N=95 (Figure 1), why 95 patients were initially considered and not 80 or 120. Does it correspond to a number of patients per unit of time in the centres of interest?

Response: After reading your comments, we have reviewed the patient enrollment and allocation. A total of 95 patients were screened, and the number represented an actual number of screened patients, which cannot be estimated. The number (n=36) represented the final eligible patients for enrollment. The determination of 36 cases is to consider the first patient trial, and to ensure statistical needs as much as possible in the event of subject detachment.

4/ 'Statistical analysis':

Some (significant) rewriting of the statistical analysis may be of benefit to the reader.

4a) MEANS:

4a-1/ lines 229 to 233: "LS means": Authors seems to be performing "tests of equality of means between groups and/or time points with Tukey's multiplicity correction" (this sounds much clearer 'LS means corrected by Tukey's method') depending on the analysis. From their description, it is unclear what analysis was actually performed.

For example

* in Table 4: are the 'LSmean (95%CI)' and the LSmean difference (95%CI) corrected for multiplicity?

* in Figures 2 and 3: how are the confidence intervals defined and are there corrected for multiplicity?

Therefore, it may be useful to list all ways of getting estimates and confidence intervals (with and without confidence intervals) and to specify in which Tables and Figures they were used.

Response: Thank you for your comment. Analysis of Covariance was used to calculate the LSmean of efficacy indicators and the LSmean difference (95% CI) corrected by Tukey's method.

LSmean (95% CI) and the LSmean difference (95%CI) in Table 4, and the data presented in Figures 4 & 5 with Tukey's multiplicity correction. As you suggested, we have made an amendment to the sentence in the revised manuscript

4a-2/ why Tukey?

Also, the authors seem to be using Tukey's HSD

(https://en.wikipedia.org/wiki/Tukey%27s_range_test) to correct for multiplicity (please confirm). This method controls for the familywise error rate when considering comparisons between *all pairs* of groups (ie, 10 comparisons in total for 4 groups). However, the authors only seem to focus on the comparison of each treatment arm

with the reference one (ie, 3 comparisons). In such a case, a Dunnett like multiplicity correction (https://en.wikipedia.org/wiki/Dunnett%27s_test) or multiplicity correction for specific contrasts of parametric models are likely to be more powerful.

Response: Thank you for your comment. Yes, considering the equal sample sizes in the drug group and the placebo group, and a higher efficiency in testing we used Tukey's test (or name Tukey's HSD) to correct for multiplicity. And a comparison between the drug group and the placebo group (3 comparisons in total) was conducted in this study.

4a-3/ model check

In small sample, the coverage of 95% confidence intervals based on the linear model *strongly* depends on the respect of linear model assumption (normality, homoscedasticity). A statement claiming that model checks were performed for each outcome and showed respect of the normality and homoscedasticity assumptions on the chosen scale (original or log scale, for example) would be useful.

Response: Thank you for your comment. Considering that, in this exploratory small-scale study of short duration, model check was not used. We appreciate your suggestion, and will keep it in mind in our future study.

4b) PROPORTIONS:

4b-1/ lines 233-234: "the Wilson method was used to calculate 95% CI": Authors seems to use Wilson score intervals to define confidence intervals for binomial proportions (<https://doi.org/10.1214/aos/1015362189> formula (3.1)). Please confirm, specify if a continuity correction was used, mention that this is for proportions, and provide a reference.

Response: Thank you for your comment, and it is confirmed that a continuity correction was not used.

4b-2/ lines 234-236: "Newcombe-Wilson score method": This seems to refer to [https://doi.org/10.1002/\(SICI\)1097-0258\(19980430\)17:8<873::AID-SIM779>3.0.CO;2-I](https://doi.org/10.1002/(SICI)1097-0258(19980430)17:8<873::AID-SIM779>3.0.CO;2-I). Please confirm, specify if a continuity correction was used and provide a reference. again, please add a comment regarding multiplicity so that it is clear to the reader when this was used.

Response: We are thankful for your comment. It is confirmed that a continuity correction was not used. In light of your comment, we have made an amendment to the sentence in the revised manuscript.

4c) OTHER: lines 237-238: The authors probably mean that the 'statistical tests' were two sided (not the p-values) and that a 5% type I error was used to reject null hypotheses.

Response: Correct! In light of your comment, we have made an amendment to the sentence in the revised manuscript.

“All the statistical tests were two sided and a 5% type I error was used to reject null hypotheses.”

5/ 'Demographics' and 'Randomisation':

Authors claim, on line 251 to 253, that demographic characteristics of the study subjects at baseline were comparable between treatment and control. Some may dispute this by pointing out that the proportion of females is (significantly, according to a Fisher test) larger in the control group (4/9) than in the treatment arms (3/27). Some may also claim that, given the statistics provided in Table 1, the patients of the control group are often older than patients in the treatment arms. It may be useful to explain how age and sex were randomised, to possibly consider a sensitivity analyses controlling for such imbalances (we recognise this may be difficult given the small sample size) and to display age (y-axis) per group (x-axis) in a plot (1 point per participant, coloured coded by sex) as supplemental material. If authors believe that such unbalances are not problematic as sex and age are not believed to be confounders, they should say so.

Response: Thank you for your thoughtful comments. We reviewed the original data of the proportion of females in the control group (4/9), which is larger than that in the treatment arms (3/27), and the patients in the control group are often older than those patients in the treatment arms. We believe that such unbalances are not problematic on the basis of our observation in the phase Ia clinical trial (Zhu et al., 2021), in which the healthy subjects were enrolled at the male-to-female ratio of 1:1 and this is a randomized placebo-controlled ascending dose study, not a parallel trial, with a small sample size in each group. Age stratification was not conducted during randomization. In light of your comment, we have rewritten the statement of demographic characteristics of the study subjects at baseline in the revised manuscript.

“Despite of the randomization design, the distribution of sex of the study subjects at baseline was unbalanced between the ZSP1601 and placebo groups (Table 1). However, such unbalances were not problematic on the basis of our previous observation in the phase Ia clinical trial, in which the healthy subjects were enrolled at the male-to-female ratio of 1:1 (Zhu et al., 2021).”

Reference

Zhu X., Wu, M., Wang, H., Li, H., Lin, J., Peng, Y., et al. Safety, tolerability, and

pharmacokinetics of the novel pan-phosphodiesterase inhibitor ZSP1601 in healthy subjects: a double-blinded, placebo-controlled first-in-human single-dose and multiple-dose escalation and food effect study. *Expert Opin Investig Drugs*. 2021; 30(5): 579-89.

6/ 'Target population':

All patients are Chinese. It may be useful to clarify if the target population is 'Asian' or if similar effects are expected on other ethnic groups (sorry if this was mentioned and if I failed to see it).

Response: Thank you very much for your comment. In this study, all patients are Chinese. With the promising results from this phase Ib/IIa trial in the Chinese patient population, we feel it is worth extending to Asian or other patient populations in future clinical trials. In light of your comment, we have made a change to “Ethnicity” in Table 1 to Asian (Chinese).

7/ 'Efficacy' section:

7a) consider splitting your efficacy section (lines 302 to 372) between primary and secondary outcomes (as defined in the protocol) by using sub sections to clearly highlight what your primary conclusions are. Indeed, the fact that you mention ALT, AST, LFC on line 303 but display ALT, AST GGT and ALP in Figure 3 but only focus on ALT and AST in Figure 4 is *confusing*. It would be helpful to display the same information for all primary outcomes (both in the different Tables and Figures).

Response: As you suggested, we have added GGT and ALP in Figure 4 to display the same information as Figure 3 (ALT, AST, GGT, and ALP). In addition, we have revised the legend for Figure 4.

“Bar graphs represented LS means with error bars reflecting two-sided 95% CIs from an Analysis of Covariance (ANCOVA) model with absolute changes in ALT (**A**), AST (**B**), GGT (**C**), and ALP (**D**) levels from baseline as the dependent variable.”

7b) lines 310: could you specify where this number came from (Table, Figure)

Response: Thank you for your suggestion. The efficacy data in Figures 3, 4, & 5, and Table 4 have been specified under the subheading “Efficacy” in Results section, while those efficacy data without being presented, such as ALT, in the Figures and Table were directly described in the main text.

7c) lines 312 to 324: if these analyses are exploratory (inspired by previous significant results and trying to explain them more deeply by 'following the data'), please say so. It is hard to track the number of tests you performed.

Response: After reading this comment and considering your comment 7a, we have made revisions and re-organization of the paragraphs under the subheading “Efficacy” in Results section. All changes were denoted in red fonts in the revised manuscript.

7d) lines 326 to 360: same as for ALT.

Response: Thank you for your suggestion. The efficacy data in Figures 3, 4, & 5, and Table 4 have been specified under the subheading “Efficacy” in Results section, while those efficacy data without being presented in the Figures and Table were directly described in the main text.

8/ 'Discussion' section:

Given your large familywise error rate, probably try to remain on the cautious side regarding your conclusions, especially when commenting on secondary outcomes.

Response: We agree and have toned down the conclusive statement and comment on secondary outcomes in Discussion section.

9/ 'Table 1':

The legend notes that 'Data are expressed as n(%), mean(SD) or median(IQR)': It is unclear when mean(SD) or median(IQR) were used. It may be useful to indicate this more precisely.

Response: We have followed your good suggestion to provide additional information about when mean (SD) or median (IQR) were used in Table 1.

10/ 'Tables 2 and 3':

* it may be useful to specify what is shown (mean(SD) or n(%) or median(IQR)) and where.

Response: As you suggested, we have made amendments to Table 2.

* in Table 3, the C max value of the first treatment group at day 1 seem to exactly equal 1. Were data standardised? If yes how?

Response: In Table 3, results were expressed as the arithmetic mean (standard deviation), except for Tmax, which was presented in the median (range). Data were not standardized.

11/ 'Table 4'

A legend in which you indicate why some results are presented with bold fonts would be useful.

Response: Your good suggestion has been well taken, and we have included the following additional information to indicate the data presented with bold fonts in Table 4.

“Bold fonts denoted statistically significant values.”

12/ 'Figure 2':

very naively, I don't seem to spot the difference between the left and right panels in Figure 2A and 2B (same titles, same x-axes, same y-axes). If this relates to a replication study, maybe consider saying so in the legend, reflecting this in the title (experiment 1, experiment 2), and using the same y-axes in both panels to make comparison easier.

Response: The left panel is linear graph while the right panel is semi-log graph. We have revised figure legend to make it clear.

13/ 'Figure 3':

* in the legend, you forgot to mention you also display GGT (C) and ALP (D).
* the confidence intervals overlap: maybe consider shifting them slightly on the x-axis to allow comparisons.

Response: Thank you for bringing this issue to our attention, and we have added the information about GGT (C) and ALP (D) in the legend for Figure 3.

With respect to the comment (the confidence intervals overlap), we reviewed the original data presented in Figure 3. In fact, the data illustrated Figure 3 were mean \pm standard deviation (SD) but not confidence intervals. After reading your comment, we realized the SD error bars overlapped, while it is hard to shift them on the x-axis because of sharing the same time point.

14/ 'Figure 4':

* your sample size per group is small enough to show all the individual data points. Please do so.
* remember to indicate in the stats section if multiplicity correction per outcome in this analysis. If so, probably have this reflected on lines 308 and 309 by preferring adj. p-value to p-value.

Response: We have presented all the individual data points in the revised Figure 4. As you suggested, we have indicated multiplicity correction for the efficacy indicators under the heading of Statistical analysis in Methods section.

15/ 'Data sharing':

Good to hear authors intend to share data. Could they maybe specify how and when?

Response: As suggested, we have specified how and when to share the data in the revised manuscript.

16/ Editing:

16a) Lines 40, 156, 158: the abbreviations QD and BID are used without explanation. On line 156 and 158, the third treatment seems to be different: '100 mg BID' on line 156, '100 mg QD' on line 158.

Response: We have spelled out when using the abbreviations QD, BID for the first time in Methods section: "including 50 mg once daily (QD), 50 mg twice daily (BID) (time interval of 12 ± 2 hr), and 100 mg BID (time interval of 12 ± 2 hr)." Correct! In light of your comment, we have made an amendment to the third treatment.

16b) Lines 39 to 41: It may be clearer to the reader to specify more clearly the different cohorts. For example "[...], were enrolled in three cohorts: (i) 50 mg once daily, (ii) 50 mg twice daily and (iii) 100 mg once daily. In each cohort, patients were randomly assigned to the drug or matching placebo with a 3:1 ratio. The primary efficacy outcome [...]". If using (i), (ii) and (iii) as suggested above on lines 39 to 41, probably do the same on lines 156 and 158.

Response: Thank you for your good suggestions, and we have specified more clearly the different cohorts and interventions under Patient allocation and interventions in Methods section.

16c) Lines 45, 131 and : 'ALT' and 'AST' are defined later on page 10 only. It would be useful to define them when first using these abbreviations

Response: Thank you for your carefulness, and we have spelled out ALT and AST when first using these two abbreviations.

16d) Line 113: Probably start a new paragraph with the word 'recently'

Response: As suggested, we have started a new paragraph with the word 'Recently'.

16e) Line 303: you seem to also consider GGT and ALP in Figure 3. Maybe add it here

Response: The interpretation of GGT and ALP has been added in the revised manuscript.

“After the treatment course, there were no significant differences in the reduction of serum ALP and GGT levels from baseline between the ZSP1601 and placebo groups (all $P > 0.05$).”

16f) Line 617-620: left and right panels instead of upper and lower ones.

Response: Thank you for bringing this issue to our attention, and we have made the following revisions as you suggested.

“Left panel, linear graph; Right panel, semi-log graph.”

Reply to Review #2:

Reviewer #2 (Remarks to the Author):

This is a phase Ib/IIa randomised trial of a pan-phosphodiesterase inhibitor, ZSP1601, in patients with nonalcoholic steatohepatitis. The study sought to evaluate the tolerability, safety, pharmacokinetics and early pharmacodynamics of the drug using a dose-escalation design, and also provides preliminary efficacy data with regards to liver biochemistry biomarker and imaging outcomes.

Response: We sincerely appreciate your recognition of the strengths of our work. We are also grateful to you for providing us with invaluable suggestions, which have been truly helpful in improving our manuscript. We have made the following corrections and explanations.

Major comments

1. The study was prospectively registered with primary outcomes relating to safety and tolerability. However, the reporting focuses on efficacy. Although the efficacy data warrants reporting, it should not be the main focus of the article and should be treated with appropriate caution given the small number of participants. The full article needs to be revised with this in mind, but in particular: the title should mention safety and tolerability; the summary should report these findings and in greater detail before mentioning the efficacy findings; and the primary and secondary outcomes should be

explicitly identified in the methods. The mention of “Primary efficacy outcomes” in the summary (Line 41) is somewhat misleading – some of these outcomes were registered as secondary outcomes, and some were not registered at all (although they are mentioned in the protocol); in the main text, they are described as exploratory. I suggest that the statement of the aims at the end of the introduction should be clearer about the main aim vs secondary aims, too.

Response: We are grateful to you for providing us with thoughtful comments and invaluable suggestions, which have been truly helpful in improving our manuscript.

We agree with your comment that the study was prospectively registered with the primary outcome relating to safety and tolerability. Although the efficacy of ZSP1601 was the secondary outcome, we believe the efficacy data of alterations from baseline in liver chemistries and LFC, and the FibroScan parameters are worth reporting. Given the small number of participants, the efficacy of ZSP1601 should not be the main focus, and should be treated with appropriate caution. Accordingly, we have revised the manuscript.

With respect to your particular comments and suggestions, we have also made changes to the manuscript, which are summarized below. Safety and tolerability have been included in the title, as suggested. We have reported the major findings of the safety and tolerability in greater detail prior to the efficacy findings in the Summary section.

In the Summary section, the primary and secondary outcomes have been explicitly identified in the methods. We have also made the information on “primary efficacy outcomes” clearer in the Summary section.

As suggested, we have made the statement of the aims clearer in the Introduction.

2. The discussion of subgroups needs to be treated with greater caution, given the very small numbers and the fact that non subgroup analyses were prespecified (as far as I can see, in the registration). When discussing subgroups, please give n, effect estimates, and CIs, not just p-values.

Response: We agree with your comment, and have provided n values, effect estimates, and CIs in the revised Summary section.

3. Please provide details of the randomisation process, including who generated the allocation sequence and the method used; the method used to implement allocation (to ensure concealment); who enrolled participants and who assigned the interventions. Please also indicate any restriction by centre – whether participants in

each dose-group were distributed across the centres, or whether each of the three centres dealt with one of the three dose-groups only.

Response: Your suggestions have been implemented, and we have included an additional paragraph describing details of patient allocation and randomization in the Methods section. Also, the subheading “Interventions” has been changed to “Patient allocation and interventions”. The additional information about patient enrollment and questions regarding exclusion criteria have been integrated into the subheading “Patients and study design” in the Methods section.

4. Methods, Safety assessment: Please describe the process for attributing adverse drug reactions.

Response: Accordingly, we have included additional information about the process used to attribute causality to suspected adverse drug reactions in the safety assessment paragraph of the Methods section.

5. Figures 4 and 5: these data would be better to displayed in dot plots to avoid information loss (see DOI: 10.1371/journal.pbio.1002128 and DOI: 10.1161/CIRCULATIONAHA.118.037777).

Response: Thank you for your comment. We have revised Figures 4 and 5 accordingly.

6. Please discuss the limitations of the study with regard to the primary safety outcome – the small number of participants and short duration means it is likely that only very common and short-term adverse reactions would be detected. Please also comment on the small number of women included – this may be insufficient to detect even common adverse reactions in women.

Response: As suggested, we have included information on the study limitations in the Discussion section.

7. The declaration of interests states: “All authors declare no conflict of interest.” However, four of the authors list their affiliation as the company that developed the drug. Please review, and be sure to declare all relevant interests that could represent a conflict (actual or perceived).

Response: After reading your comment, we realized that authors who are employees of the drug company should declare a conflict of interest. As such, we have revised the Declaration of Interests.

8. In some places, the language, although grammatically correct, is either ambiguous

in meaning or reports something other than what I think the authors are trying to say, specifically:

- Lines 38-41: The description of the trial arms in the summary is unclear – I would think (and the protocol seems to indicate) that group 1 would comprise 9 participants on 50 mg QD and 3 on placebo; then the second arm would introduce the next dose (9 participants on 50 mg BID, 3 on placebo) started after a time-lag, etc – but the summary implies each cohort of 12 had 3 participants on each dose + 3 on placebo, and methods section is ambiguous. Please clarify this. It might be helpful to mention the criteria for instigating the next dose-arm. A figure similar to the dose escalation flowchart from the protocol would be helpful, either as a figure in the main text, or a supplementary file.

Response: Thank you for bringing this issue to our attention. In light of this suggestion, we have revised the description of the trial arms in the Summary section, to make the information clearer.

As suggested, we have included a dose escalation flowchart in the revised Figure 1.

- Line 44: meaning of “trend” is unclear here. If you’re trying to say that there were non-significant differences, it would be better to talk about observed difference but acknowledge uncertainty in the estimates and that it could be consistent with random variation.

Response: We agree with your comment on the lack of clarity in the meaning of the word “trend”. As the Summary section ended up being too long, we have decided to delete the sentence.

- Line 51: “Exposure to ZSP1601 and its metabolites showed dose-proportional increases.” – is this still talking about adverse events, or pharmacokinetics? (increases in what?)

Response: Thank you for catching this missing information, we have added “in pharmacokinetic” in the sentence.

- Line 166: “Liver biochemical tests included...” – “included” implies that there were other liver biochemical tests conducted that are not listed. Please list all tests (and similarly for biomarkers).

Response: We agree with your comment and have revised the sentence.

- Lines 318-322: “When stratified by BMI, a significant reduction of serum ALT levels was observed in both ZSP1601 50 mg BID and 100 mg BID groups when compared to the placebo group (P=0.0002 and P=0.0011) in patients with a BMI \geq 28 kg/m².” – “a significant reduction” implies a null hypothesis of no reduction between baseline and post-drug. “compared to the placebo group” implies the null hypothesis is no difference in the size of the reduction between experimental and placebo drugs. Please clarify which is being assessed – either, “there was a significant reduction in [variable] in groups x and y but not in group z”, or “there was a significantly larger reduction in [variable] in both group x and group y compared with in group z”. I think it’s the latter (which is the more appropriate comparison, by my understanding).

Response: Thank you for your thoughtful comment and helpful suggestion. Your understanding is correct. We have clarified the interpretation of the results in the revised manuscript.

- Similarly, Line 329- “Compared with patients who received placebo, AST level were significantly decreased from baseline...” I think you mean, “compared with in patients who received placebo, AST levels showed a significantly greater decrease from baseline in patients treated with...[active drug]”

Response: Correct! Thank you very much for the suggested revision to the sentence, we have done so in the revised manuscript.

- Line 468-474: The data sharing statement is unclear. “All data will be reviewed...” – do you mean that all requests for data access will be reviewed and evaluated as described? Please indicate to whom requests for access should be directed.

Response: After reading your comment, we have rewritten this section.

Minor comments

1. The “reporting summary” indicates that a completed CONSORT checklist should be provided with the submission, but I could not find this in the materials provided. Please supply a CONSORT checklist, ensuring all items are fully reported.

Response: As suggested, we have supplied a CONSORT checklist in the revised manuscript submission.

2. Please mention in the title that this is a randomized, placebo-controlled phase Ib/IIa trial, for ease of searching and data extraction.

Response: Thank you for your suggestion, we have revised the title to mention that

this is a randomized, placebo-controlled phase Ib/IIa trial in the revised manuscript.

3. Lines 112-113: “In preclinical studies, ZSP1601 demonstrated greater therapeutic effectiveness in comparison with PTX” – needs a citation (also, what is “therapeutic” meaning here, given that this is referring to a preclinical study?)

Response: Thanks you for your comment. We have added the in-text citation and made a minor revision to the statement.

Reference

Zhu X., Wu, M., Wang, H., Li, H., Lin, J., Peng, Y., et al. Safety, tolerability, and pharmacokinetics of the novel pan-phosphodiesterase inhibitor ZSP1601 in healthy subjects: a double-blinded, placebo-controlled first-in-human single-dose and multiple-dose escalation and food effect study. *Expert Opin Investig Drugs*. 2021; 30(5): 579-89.

4. Lines 119-121: In establishing the need for your study, you state that: “further clinical trials to evaluate the efficacy of ZSP1601 in the treatment of NASH” are warranted – but the primary outcome of this study was safety, not efficacy. I’d suggest mentioning/justifying needing to evaluate safety and PK in patients with NASH (as opposed to healthy volunteers).

Response: Thank you for your comment and suggestion. We have revised the sentence.

5. Line 133-134: “fatty liver as confirmed by abdominal ultrasound” -what were the criteria for this?

Response: Thank you for your question. Abdominal ultrasound was used as an auxiliary examination shows fatty infiltration of the liver based on ultrasound findings. Accordingly, we have improved the description in the revised manuscript to make this information clearer.

6. Line 141: “et al.” is vague – please specify all exclusion criteria

Response: We agree that “et al.” is vague and have made it clearer in the revised manuscript. As there is a relatively long list of chronic liver diseases, it would be better to use “chronic liver diseases other than NASH” and then split into two criteria as follows:

“(1) history or presence of chronic liver diseases other than NASH; (2) cirrhosis, hepatic decompensation, or HCC;”

7. Line 147: please provide the ethics approval number(s)

Response: We have followed your suggestion and provided the ethics approval number in the revised manuscript.

“(Approval numbers: 19Y110-007, NFEC-201909-Y4, 2 019-PI-036-04)”

8. Interventions: Please indicate the time-lag of ascending doses and the criteria for instigating (or not instigating) the next dose-group

Response: Thank you for your comment. We have included the dose escalation flowchart in the revised Figure 1.

9. Interventions: Please indicate whether and how compliance was monitored

Response: In this study, patient compliance was monitored. This information has been added to the revised manuscript.

10. Line 189: Please cite or describe the tool used for documentation

Response: As you suggested, we have described the tool used for documentation in the revised manuscript.

11. Line 194: “as reported previously” – please provide a citation.

Response: After reading your comment, we have made this sentence clearer in the revised manuscript.

12. Line 201: “including” – please list all

Response: Thank you for your comment. We have revised this sentence.

13. Line 215: “etc” – please list all

Response: Thank you for your comment, we have made revisions to the sentence.

14. Please justify dichotomising ALT, and indicate the basis for choosing $\geq 30\%$ reduction from baseline as an outcome.

Response: Thank you for your comment. Based on the Farnesoid X Receptor Ligand

Obeticholic Acid in NASH Treatment (FLINT) clinical trial study for NASH that showed a decrease in ALT greater than 17 U/L could be used to predict histological improvement or resolution of NASH, a $\geq 30\%$ reduction from baseline can meet a decrease in ALT greater than 17 U/L. Therefore we choose a $\geq 30\%$ reduction from baseline as an outcome.

15. Table 2: If n is the number of events, as indicated in the footnote, then the numbers do not add up. Possibly it is the number of participants experiencing one or more event?

Response: “n” is the number of subjects who experienced at least one AE. We have revised the table to make this information clearer.

16. Figures 3 and 5: please indicate in the figure legend what the error bars represent.

Response: In Figures 3 and 5, the error bars represent standard deviation values. As suggested, we have included this information in the figure legends.

17. Line 315: “There was an obviously dose-dependent tendency” – maybe “apparent” or “possible” dose-dependence would be better, given the small numbers and lack of statistical significance (and low power). (And “tendency” should be removed).

Response: We have followed your suggestion and made revisions to this sentence.

18. Line 332-335: “The proportion of patients with reduced AST levels $\geq 30\%$ from baseline was greater in the ZSP1601 100 mg BID group compared with the placebo group, whereas there was no significant difference between ZSP1601 treatment groups and placebo groups.” – do you mean no significant difference between the 50 mg QD or the 50 mg BID group and the pooled placebo group? Also, please don’t just say “not significant”, please give the numbers and p-value.

Response: After reading your comments, we have revised the sentence. In addition, the numbers have been included in the revised sentence.

19. Paragraph beginning line 337, “Alterations in other liver chemistries....” – too much focus on significant/non-significant dichotomy – better to discuss the scale of the difference and precision of estimate.

Response: Thank you for your comment, we have revised the sentence.

20. Lines 345-347: “[LS mean (95% CI), -2.49(-5.52 to 0.54), -2.89(-5.94 to 0.16),

-4.98(-7.62 to -2.33) and 347 -5.42(-8.64 to -2.19) in placebo and ZSP1601 groups, respectively"-please specify which ZSP1601 group each value refers to.

Response: As suggested, we have revised the sentence.

21. Lines 391-392: "there was no significant difference in safety profiles between ZSP1601 and placebo groups" – unless I'm missing something, you didn't apply any statistical tests here, so please change the wording to avoid the term "significant".

Response: In light of your comment and considering the safety and tolerability data presented in the Results section, we feel it would be better to remove it from this information from the sentence.

22. Lines 433-435: "Firstly, the number of enrolled patients is relatively small, which may lead to less statistical significance between the ZSP1601 and the placebo groups." – This is a rather upside-down way of looking at it, and suggests the only concern is hitting " $p < 0.05$ " rather than evaluating the strength of the evidence for a clinically meaningful effect. The small sample size limits the power of the study to detect differences, and means the efficacy results need to be interpreted with caution.

Response: Thank you for your helpful information, we have revised this sentence in the Discussion section.

23. Please indicate the role of the funders (or lack thereof) in the study design, data collection, analysis, decision to publish, and manuscript preparation.

Response: The funders had no role in the study design, data collection, data analysis, decision to publish, and manuscript preparation.

Again, we sincerely appreciate your insightful comments and valuable suggestions. Hopefully, we have addressed your comments. If you have any remaining concerns or suggestions, we will make amendments to the manuscript under your guidance.

Reply to Review #3:

Reviewer #3 (Remarks to the Author):

This manuscript focus on the ZSP1601 therapy for NASH patients. The manuscript is easier to follow, the design of the experiments is adequate and the experiments are executed correctly, but it requires the following corrections and explanations:

Response: We sincerely appreciate your recognition of the strengths of our work. We are also grateful to you for providing us with invaluable suggestions, which have been truly helpful in improving our manuscript. We have made the following corrections and explanations.

Point 1. Line 153, ZSP1601 tablets are 25mg and 100mg, or are they 50mg and 100mg?

Response: Thank you for your question. Two dosage forms of ZSP1601 tablets are 25mg and 100mg in this study, and patients in the ZSP1601 50 mg QD group needed to take 2 tables once daily. We have included information regarding the two dosage forms in the revised manuscript.

Point 2. Line 165 correct typo.

Response: We are impressed by your carefulness, and the typo has been corrected.

Point 3. Clarify why 14 days after treatment is a good time point.

Response: This study was built upon the phase 1a trial of ZSP1601, in which healthy subjects were continuously administered ZSP1601 or placebo for 14 days. The phase 1a study showed that the concentrations on the 7th and 14th days were slightly higher than those on the 1st day, whereas the drug time curves on the 7th and 14th days almost overlapped, indicating that the drug had reached a stable state after 7 days of continuous administration. However, based on the safety data of healthy subjects and after considering abnormal liver function in NASH patients, for safety reasons, the observation was set until the 14th day of administration for dose escalation in this study.

Point 4. The authors do not mention in Safety and tolerability results the AE≥3 (severe).

Response: Thank you for your comments. In this study, 4 patients experienced grade 3 or 4 hypertriglyceridemia in the placebo group, while 2 patients experienced grade 3 hypertriglyceridemia in the ZSP1601 50mg QD group. However, all grade ≥3 AEs were neither drug-related nor SAE. In light of this comment, we have made amendments to the Results section in the revised manuscript.

Point 5. In discussion section, the authors indicate statistically significant reductions in markers of inflammation; explain this point and the results that prove this conclusion.

Response: After reading your comment, we reviewed our original data. We examined serum inflammatory cytokines, including TNF- α , IL-6, IL-10, and IL-22, and there were no significant differences between the placebo and ZSP1601 treatment groups at the end of treatment, compared to baseline. Given the above observation and considering ALT/AST as the main evaluation of NASH in this study, we feel it would be better to delete the statement of statistically significant reductions in markers of inflammation in the Discussion and Summary sections, and have removed “markers of inflammation”. However, if you have any further concerns or suggestions regarding this issue, we shall make further changes to the manuscript. We appreciate your understanding.

Point 6. Explain the CAP meaning.

Response: As suggested, we have explained the meaning of CAP under the Laboratory and clinical examinations subheading.

Point 7. Line 624 AST, aspartate aminotransferase.

Response: Thank you for catching this issue, it has been corrected.

Point 8. Figure 3. C and D are missing in the figure legend.

Response: Thank you for bringing this issue to our attention, we have made amendments to the figure legend.

Point 9. Figure 4. Correct the letters of the figures, and explain what ANCOVA is in the statistics methods.

Response: Accordingly, we have corrected the letters in Figure 4 and added an explanation of ANCOVA in the statistical methods.

Point10. Figure 2. Indicate the figures differences (linear and semi log in the figures), they seem the same graph but with the Y-axis changed.

Response: Thank you for your suggestion, we have indicated the differences in the figure legend: Left panel, linear graph; Right panel, semi-log graph.

As always, we sincerely appreciate your comments and suggestions. If you have any remaining concerns, we shall make amendments under your guidance.

Reply to Review #4:

Reviewer #4 (Remarks to the Author):

The authors performed a 28 day study of ZSP1601 in 36 subjects (3 cohorts of 12 subjects each at 3 different escalating doses of drug) with NAFLD. Safety, tolerability and key non-invasive tests to include liver chemistry tests and liver fat content as measured by both CAP and MRI-PDFF were assessed. The authors demonstrated in this short trial with a small number of subjects that the drug was generally well tolerated and there was a dose response to ALT, AST and liver fat content. This is a very small, proof of concept study with a novel mechanism in NAFLD subjects. The main concern with the data is the small # of subjects (only 3 pbo subjects and 9 treated subjects in each cohort). Additionally, treatment was limited to 28 days.

Response: We are grateful for your recognition of the strengths of our study. We would like to thank you very much for providing us with thoughtful comments and suggestions, which have helped us in improving the manuscript. As you indicated, the study has limitations, such as a small sample size and short-term treatment, which have been identified in the Discussion section. Future clinical trials with larger sample sizes and longer-term treatments are needed to verify the safety and efficacy of ZSP1601.

With respect to your additional comments, we have made the following responses.

I have several additional comments relative to this study

1. The authors should narrow the focus of the background to their specific mechanism of action and minimize discussion of other mechanisms such as FXR, THR-beta, vitamin E, etc. In addition, there should be an explanation of what is expected to improve by working through this mechanism. While I understand you are looking at ALT as a marker of inflammation, why MRI-PDFF? What rationale is there to assess liver fat content with your mechanism? Do you impact fibrosis? This should be mentioned as well.

Response: In light of the good suggestions, we have narrowed the focus of the background. In addition, we have also provided a rationale for MRI-PDFF and fat content in the revised manuscript. We also present the additional references below for your convenience:

References

Ajmera et al. Magnetic Resonance Imaging Proton Density Fat Fraction Associates With Progression of Fibrosis in Patients With Nonalcoholic Fatty Liver Disease. *Gastroenterology*. 2018; 155(2): 307–310.e2.

Park CC, et al. Magnetic Resonance Elastography vs Transient Elastography in Detection of Fibrosis and Noninvasive Measurement of Steatosis in Patients With Biopsy-Proven Nonalcoholic Fatty Liver Disease. *Gastroenterology*. 2017; 152:598–607.e2.

Le TA, et al. Effect of colesevelam on liver fat quantified by magnetic resonance in nonalcoholic steatohepatitis: a randomized controlled trial. *Hepatology*. 2012; 56:922–32.

Loomba R, et al. Ezetimibe for the treatment of nonalcoholic steatohepatitis: assessment by novel magnetic resonance imaging and magnetic resonance elastography in a randomized trial (MOZART trial). *Hepatology*. 2015; 61:1239–50.

2. The title is not accurate. You should replace "NASH" in the title with "NAFLD" as this is not a biopsy proven NASH cohort. In fact, very little is done non-invasively to establish the diagnosis of NASH outside of elevated ALT, BMI \geq 25 kg/m² and liver fat by imaging.

Response: Thank you for your comment. Although liver biopsies were not used to make a diagnosis of NASH in this study, the corresponding manifestations are consistent with clinical diagnosis. According to the relevant guidelines in the early proof-of-concept clinical trial, biochemical criteria in combination with imaging evidence, such as lipodegeneration/steatohepatitis/fibrosis, can be used to select patients.

3. Define excessive alcohol use.

Response: We have followed your suggestion and defined excessive alcohol use in this study.

4. What defines a "normal" ALT?

Response: In this study, normal ALT levels are < 50 units per liter (U/L) for males, and < 40 U/L for females.

5. Figure one is your consort diagram. It would be nice to also include a study design figure.

Response: After considering your comment and that of another reviewer, we have integrated the dose escalation flowchart into the original figure 1. As a result, the revised Figure 1 will also reflect the study design.

6. While this is a 28 day study, did any patients lose weight and if so, is this accounted for in the non-invasive test assessments?

Response: Thank you for your question. Subjects with weight loss plans were excluded from this study. There were no lifestyle interventions, and a placebo control was included in this study. There was no statistical difference between the test group and the control group in terms of non-invasive test assessments such as waist circumference, abdominal circumference and body weight.

7. Please do not use "liver cirrhosis" together. This is an oxymoron.

Response: As suggested, we revised this throughout the manuscript.

8. Please have your paper reviewed for English grammar.

Response: The revised manuscript has been edited and proofread by a scientific editor from *Medjaden Inc.*.

9. The discussion should include commentary of next steps, including next phase studies and is there sufficient tox data to support the trial duration.

Response: Your suggestion has been taken into consideration and we have including information on next steps in the Discussion section.

Again, thank you so much for your thoughtful comments and suggestions. Hopefully we have addressed your concerns and the revised manuscript will be acceptable. If you have any remaining concerns, we shall make amendments under your guidance.

Point-by-Point Responses: NCOMMS-22-49299A

Reviewer #1 comments to (some) responses

Reply to Review #1:

Dear Editor,

As a statistician, I mainly focused on the quantitative aspects of the manuscript.

Comments/questions:

1/ 'Multi-dose escalation' & 'ascending doses':

These concepts, used on lines 127, 155, 232, 475, 427, 449 and 551, usually refer to adaptive trials in which allocation to the dose of the cth cohort depends on data collected on cohorts 1 to c-1 (like seemingly considered in the phase I trial mentioned on line 115). It doesn't seem to be the case this phase Ib/IIa. Please clarify or prefer 'multiple-dose trial' without mentioning 'escalation' or 'ascending'.

Response: We are grateful to you for providing us with thoughtful comments and suggestions, which have helped us in improving our manuscript. With respect to the issues on the quantitative aspects, we have made the following amendments or explanations.

I thank the authors for sharing their raw data with reviewers during this revision process: this is **very much appreciated**.

Actually, this is an ascending dose study and we have made it clearer in Figure 1. In addition, we have checked the entire manuscript and made revisions as well.

I disagree. 'Multi-dose escalation' or 'ascending doses' typically implies the use of a design in which a **stepwise approach** starting with a low dose and gradually increasing the dosage level (according to some rules) is used. This is not the case here where you consider different doses in a parallel design. I therefore stick to my suggestion of "multi-dose trial" (which implies a set of increasing doses when ranked by dose levels). This position is shared by a few colleagues working on dose-finding.

2/ 'Multiplicity':

Between lines 167 and 222, more than 50 outcomes are described. A statement regarding multiplicity correction for primary and secondary outcomes in the 'statistical analysis' section would be welcomed.

For example, in Table 4, the mean of each treatment group is compared to the reference one for ~15 outcomes, *possibly* (unclear) with a multiplicity correction at the outcome level. This still leaves a rather high chance of detecting differences when there is none (like for CAP and FAST, for example). Therefore, a factual statement

warning the reader of the 'large' chance of false positives in secondary outcomes would be useful for the reader to take the statements of lines 362-363 (outcome CAP) and 365-366 (outcome FAST) with caution.

Response: Thank you for your comment. We performed multiplicity correction for all the efficacy indicators (secondary outcomes) were presented in figures, tables and source data. In light of your comment and suggestion, we have included the additional information about the multiplicity correction under the heading of Statistical analysis in Methods section.

Thanks.

3/ 'Sample size':

Phase Ib/IIa trials typically consider small sample sizes. Could the authors still mention why n=36 (Figure 1) was preferred to n=24 or n=45 for example, or, if they started with N=95 (Figure 1), why 95 patients were initially considered and not 80 or 120. Does it correspond to a number of patients per unit of time in the centres of interest?

Response: After reading your comments, we have reviewed the patient enrollment and allocation. A total of 95 patients were screened, and the number represented an actual number of screened patients, which cannot be estimated. The number (n=36) represented the final eligible patients for enrollment. The determination of 36 cases is to consider the first patient trial, and to ensure statistical needs as much as possible in the event of subject detachment.

Thanks. The question remains unanswered though: why 95 and not, say, 80 or 120?

4/ 'Statistical analysis':

Some (significant) rewriting of the statistical analysis may be of benefit to the reader.

4a) MEANS:

4a-1/ lines 229 to 233: "LS means": Authors seems to be performing "tests of equality of means between groups and/or time points with Tukey's multiplicity correction" (this sounds much clearer 'LS means corrected by Tukey's method') depending on the analysis. From their description, it is unclear what analysis was actually performed. For example

* in Table 4: are the 'LSmean (95%CI)' and the LSmean difference (95%CI) corrected for multiplicity?

* in Figures 2 and 3: how are the confidence intervals defined and are there corrected for multiplicity?

Therefore, it may be useful to list all ways of getting estimates and confidence intervals (with and without confidence intervals) and to specify in which Tables and

Figures they were used.

Response: Thank you for your comment. Analysis of Covariance was used to calculate the LSmean of efficacy indicators and the LSmean difference (95% CI) corrected by Tukey's method.

LSmean (95% CI) and the LSmean difference (95%CI) in Table 4, and the data presented in Figures 4 & 5 with Tukey's multiplicity correction. As you suggested, we have made an amendment to the sentence in the revised manuscript

I am again **very grateful** to the authors to have shared their raw data as it allowed to just check if I can reproduce randomly chosen results.

In the section efficacy, I can reproduce the results of lines 339-340 (relative decrease in ALT levels) and 346, but not the absolute decrease levels of line 338 which makes me wonder how they were defined. For each patient, I defined the difference between ALT at day 29 minus ALT at baseline and modelled this difference as a function of dose level (ie, outcome = difference in ALT level, dose level as only predictor). Did you use a different model?

I did the same for AST and can reproduce the results of lines 357-358 for all conditions but 50 mg BID (manuscript indicates 28.76% while I get 31.7%) and don't seem able to reproduce the results related to the absolute AST difference of lines 361-362, likely for the same reason as above.

4a-2/ why Tukey?

Also, the authors seem to be using Tukey's HSD (https://en.wikipedia.org/wiki/Tukey%27s_range_test) to correct for multiplicity (please confirm). This method controls for the familywise error rate when considering comparisons between *all pairs* of groups (ie, 10 comparisons in total for 4 groups). However, the authors only seem to focus on the comparison of each treatment arm with the reference one (ie, 3 comparisons). In such a case, a Dunnett like multiplicity correction (https://en.wikipedia.org/wiki/Dunnett%27s_test) or multiplicity correction for specific contrasts of parametric models are likely to be more powerful.

Response: Thank you for your comment. Yes, considering the equal sample sizes in the drug group and the placebo group, and a higher efficiency in testing we used Tukey's test (or name Tukey's HSD) to correct for multiplicity. And a comparison between the drug group and the placebo group (3 comparisons in total) was conducted in this study.

Thanks. I seem not to have been able to share my point as well as wanted. You seem

to have picked a multiplicity correction considering all pairwise comparisons (Tuckey's HSD) while being mainly interested in only 3 of them (3 treatments versus control). In this case, a Dunnett like multiplicity correction leads to a smaller multiplicity correction related power loss. **As your approach is likely more conservative than necessary, all good.**

4a-3/ model check

In small sample, the coverage of 95% confidence intervals based on the linear model *strongly* depends on the respect of linear model assumption (normality, homoscedasticity). A statement claiming that model checks were performed for each outcome and showed respect of the normality and homoscedasticity assumptions on the chosen scale (original or log scale, for example) would be useful.

Response: Thank you for your comment. Considering that, in this exploratory small-scale study of short duration, model check was not used.

From a statistician point of view, your answer is golden. You draw conclusions based on the use of statistical techniques. These conclusions may not be valid if the related assumptions are not met **especially** when the sample size is too small to rely on the central limit theorem, i.e., **especially** for small-scale studies.

We appreciate your suggestion, and will keep it in mind in our future study.

I strongly suggest that you perform model checks, at least for all analyses leading to statistical significance (ie, focusing on potential false positive). For efficacy outcomes, this means checking that the conditional normality and homoscedasticity assumptions are met so that you can claim that "*model checks showed a good fit of the model to the data*" in the manuscript. This will reassure this reviewer and a few of your readers. **Model check is not an option with small sample sizes.**

4b) PROPORTIONS:

4b-1/ lines 233-234: "the Wilson method was used to calculate 95% CI": Authors seems to use Wilson score intervals to define confidence intervals for binomial proportions (<https://doi.org/10.1214/aos/1015362189> formula (3.1)). Please confirm, specify if a continuity correction was used, mention that this is for proportions, and provide a reference.

Response: Thank you for your comment, and it is confirmed that a continuity correction was not used.

Thanks.

4b-2/ lines 234-236: "Newcombe-Wilson score method": This seems to refer to [https://doi.org/10.1002/\(SICI\)1097-0258\(19980430\)17:8<873::AID-SIM779>3.0.CO;2-I](https://doi.org/10.1002/(SICI)1097-0258(19980430)17:8<873::AID-SIM779>3.0.CO;2-I) . Please confirm, specify if a continuity correction was used and provide a reference. again, please add a comment regarding multiplicity so that it is clear to the reader when this was used.

Response: We are thankful for your comment. It is confirmed that a continuity correction was not used. In light of your comment, we have made an amendment to the sentence in the revised manuscript.

Thanks.

4c) OTHER: lines 237-238: The authors probably mean that the 'statistical tests' were two sided (not the p-values) and that a 5% type I error was used to reject null hypotheses.

Response: Correct! In light of your comment, we have made an amendment to the sentence in the revised manuscript.

“All the statistical tests were two sided and a 5% type I error was used to reject null hypotheses.”

Perfect.

5/ 'Demographics' and 'Randomisation':

Authors claim, on line 251 to 253, that demographic characteristics of the study subjects at baseline were comparable between treatment and control. Some may dispute this by pointing out that the proportion of females is (significantly, according to a Fisher test) larger in the control group (4/9) than in the treatment arms (3/27). Some may also claim that, given the statistics provided in Table 1, the patients of the control group are often older than patients in the treatment arms. It may be useful to explain how age and sex were randomised, to possibly consider a sensitivity analyses controlling for such imbalances (we recognise this may be difficult given the small sample size) and to display age (y-axis) per group (x-axis) in a plot (1 point per participant, coloured coded by sex) as supplemental material. If authors believe that such unbalances are not problematic as sex and age are not believed to be confounders, they should say so.

Response: Thank you for your thoughtful comments. We reviewed the original data of the proportion of females in the control group (4/9), which is larger than that in the treatment arms (3/27), and the patients in the control group are often older than those

patients in the treatment arms.

Thanks. I think it is important to mention the fact that the randomisation led to some level of imbalance in the sex ratio and average age between conditions.

Thanks to the authors sharing their raw data, I could note that the average age is indeed significantly higher in the control group (the higher the dose, the lower the age). This may lead to a potential confounding when authors detect a relationship between an outcome and the dose level: **is it due to the dose difference or to the age or sex?**

We believe that such unbalances are not problematic on the basis of our observation in the phase Ia clinical trial (Zhu et al., 2021), in which the healthy subjects were enrolled at the male-to-female ratio of 1:1 and this is a randomized placebo-controlled ascending dose study, not a parallel trial, with a small sample size in each group. Age stratification was not conducted during randomization. In light of your comment, we have rewritten the statement of demographic characteristics of the study subjects at baseline in the revised manuscript.

I am afraid I don't think that this argument (mentioned on lines 279-281) is very convincing. If authors want to claim that their conclusions are valid while facing some level of imbalance in the sex ratio and average age between conditions, they should explain that sex and age have a marginal effect on the outcome of interest given the dose (or an effect going on the opposite direction of the dose effect).

Again, authors may be willing to check that their conclusions hold while controlling for age and/or sex which seems pretty straightforward in the ANCOVA framework they explained using above. They could then either [i] mention in the manuscript that "Sensitivity analyses controlling for age led to the same conclusions" which would reassure this reviewer and some readers or [ii] explain that causality is hard to define due to potential confounding in specific cases.

"Despite of the randomization design, the distribution of sex of the study subjects at baseline was unbalanced between the ZSP1601 and placebo groups (Table 1). However, such unbalances were not problematic on the basis of our previous observation in the phase Ia clinical trial, in which the healthy subjects were enrolled at the male-to-female ratio of 1:1 (Zhu et al., 2021)."

Reference

Zhu X., Wu, M., Wang, H., Li, H., Lin, J., Peng, Y., et al. Safety, tolerability, and pharmacokinetics of the novel pan-phosphodiesterase inhibitor ZSP1601 in healthy

subjects: a double-blinded, placebo-controlled first-in-human single-dose and multiple-dose escalation and food effect study. Expert Opin Investig Drugs. 2021; 30(5): 579-89.

6/ 'Target population':

All patients are Chinese. It may be useful to clarify if the target population is 'Asian' or if similar effects are expected on other ethnic groups (sorry if this was mentioned and if I failed to see it).

Response: Thank you very much for your comment. In this study, all patients are Chinese. With the promising results from this phase Ib/IIa trial in the Chinese patient population, we feel it is worth extending to Asian or other patient populations in future clinical trials. In light of your comment, we have made a change to “Ethnicity” in Table 1 to Asian (Chinese).

Thanks.

7/ 'Efficacy' section:

7a) consider splitting your efficacy section (lines 302 to 372) between primary and secondary outcomes (as defined in the protocol) by using sub sections to clearly highlight what your primary conclusions are. Indeed, the fact that you mention ALT, AST, LFC on line 303 but display ALT, AST GGT and ALP in Figure 3 but only focus on ALT and AST in Figure 4 is *confusing*. It would be helpful to display the same information for all primary outcomes (both in the different Tables and Figures).

Response: As you suggested, we have added GGT and ALP in Figure 4 to display the same information as Figure 3 (ALT, AST, GGT, and ALP). In addition, we have revised the legend for Figure 4.

“Bar graphs represented LS means with error bars reflecting two-sided 95% CIs from an Analysis of Covariance (ANCOVA) model with absolute changes in ALT **(A)**, AST **(B)**, GGT **(C)**, and ALP **(D)** levels from baseline as the dependent variable.”

7b) lines 310: could you specify where this number come from (Table, Figure)

Response: Thank you for your suggestion. The efficacy data in Figures 3, 4, & 5, and Table 4 have been specified under the subheading “Efficacy” in Results section, while those efficacy data without being presented, such as ALT, in the Figures and Table were directly described in the main text.

7c) lines 312 to 324: if these analyses are exploratory (inspired by previous significant results and trying to explain them more deeply by 'following the data'), please say so.

It is hard to track the number of tests you performed.

Response: After reading this comment and considering your comment 7a, we have made revisions and re-organization of the paragraphs under the subheading “Efficacy” in Results section. All changes were denoted in red fonts in the revised manuscript.

7d) lines 326 to 360: same as for ALT.

Response: Thank you for your suggestion. The efficacy data in Figures 3, 4, & 5, and Table 4 have been specified under the subheading “Efficacy” in Results section, while those efficacy data without being presented in the Figures and Table were directly described in the main text.

8/ 'Discussion' section:

Given your large familywise error rate, probably try to remain on the cautious side regarding your conclusions, especially when commenting on secondary outcomes.

Response: We agree and have toned down the conclusive statement and comment on secondary outcomes in Discussion section.

9/ 'Table 1':

The legend notes that 'Data are expressed as n(%), mean(SD) or median(IQR)': It is unclear when mean(SD) or median(IQR) were used. It may be useful to indicate this more precisely.

Response: We have followed your good suggestion to provide additional information about when mean (SD) or median (IQR) were used in Table 1.

10/ 'Tables 2 and 3':

* it may be useful to specify what is shown (mean(SD) or n(%) or median(IQR)) and where.

Response: As you suggested, we have made amendments to Table 2.

* in Table 3, the C max value of the first treatment group at day 1 seem to exactly equal 1. Were data standardised? If yes how?

Response: In Table 3, results were expressed as the arithmetic mean (standard deviation), except for Tmax, which was presented in the median (range). Data were not standardized.

11/ 'Table 4'

A legend in which you indicate why some results are presented with bold fonts would be useful.

Response: Your good suggestion has been well taken, and we have included the following additional information to indicate the data presented with bold fonts in Table 4.

“Bold fonts denoted statistically significant values.”

12/ 'Figure 2':

very naively, I don't seem to spot the difference between the left and right panels in Figure 2A and 2B (same titles, same x-axes, same y-axes). If this relates to a replication study, maybe consider saying so in the legend, reflecting this in the title (experiment 1, experiment 2), and using the same y-axes in both panels to make comparison easier.

Response: The left panel is linear graph while the right panel is semi-log graph. We have revised figure legend to make it clear.

13/ 'Figure 3':

- * in the legend, you forgot to mention you also display GGT (C) and ALP (D).
- * the confidence intervals overlap: maybe consider shifting them slightly on the x-axis to allow comparisons.

Response: Thank you for bringing this issue to our attention, and we have added the information about GGT (C) and ALP (D) in the legend for Figure 3.

With respect to the comment (the confidence intervals overlap), we reviewed the original data presented in Figure 3. In fact, the data illustrated Figure 3 were mean \pm standard deviation (SD) but not confidence intervals. After reading your comment, we realized the SD error bars overlapped, while it is hard to shift them on the x-axis because of sharing the same time point.

Basing the plot on an amended dataset I which very slightly shift the time points per condition (+/- 0.1, for example) may be useful to avoid such overlaps

14/ 'Figure 4':

- * your sample size per group is small enough to show all the individual data points.

Please do so.

* remember to indicate in the stats section if multiplicity correction per outcome in this analysis. If so, probably have this reflected on lines 308 and 309 by preferring adj. p-value to p-value.

Response: We have presented all the individual data points in the revised Figure 4. As you suggested, we have indicated multiplicity correction for the efficacy indicators under the heading of Statistical analysis in Methods section.

Thanks. As above, basing the plot on an amended dataset I which you very slightly shift the points on the x-axis may be useful to avoid overlaps of points.

15/ 'Data sharing':

Good to hear authors intend to share data. Could they maybe specify how and when?

Response: As suggested, we have specified how and when to share the data in the revised manuscript.

16/ Editing:

16a) Lines 40, 156, 158: the abbreviations QD and BID are used without explanation. On line 156 and 158, the third treatment seems to be different: '100 mg BID' on line 156, '100 mg QD' on line 158.

Response: We have spelled out when using the abbreviations QD, BID for the first time in Methods section: "including 50 mg once daily (QD), 50 mg twice daily (BID) (time interval of 12 ± 2 hr), and 100 mg BID (time interval of 12 ± 2 hr)." Correct! In light of your comment, we have made an amendment to the third treatment.

16b) Lines 39 to 41: It may be clearer to the reader to specify more clearly the different cohorts. For example "[...], were enrolled in three cohorts: (i) 50 mg once daily, (ii) 50 mg twice daily and (iii) 100 mg once daily. In each cohort, patients were randomly assigned to the drug or matching placebo with a 3:1 ratio. The primary efficacy outcome [...]" If using (i), (ii) and (iii) as suggested above on lines 39 to 41, probably do the same on lines 156 and 158.

Response: Thank you for your good suggestions, and we have specified more clearly the different cohorts and interventions under Patient allocation and interventions in Methods section.

16c) Lines 45, 131 and : 'ALT' and 'AST' are defined later on page 10 only. It would be useful to define them when first using these abbreviations

Response: Thank you for your carefulness, and we have spelled out ALT and AST when first using these two abbreviations.

16d) Line 113: Probably start a new paragraph with the word 'recently'

Response: As suggested, we have started a new paragraph with the word 'Recently'.

16e) Line 303: you seem to also consider GGT and ALP in Figure 3. Maybe add it here

Response: The interpretation of GGT and ALP has been added in the revised manuscript.

“After the treatment course, there were no significant differences in the reduction of serum ALP and GGT levels from baseline between the ZSP1601 and placebo groups (all $P > 0.05$).”

16f) Line 617-620: left and right panels instead of upper and lower ones.

Response: Thank you for bringing this issue to our attention, and we have made the following revisions as you suggested.

“Left panel, linear graph; Right panel, semi-log graph.”

REVIEWER COMMENTS

Reviewer #2 (Remarks to the Author):

The authors have put considerable work into revising the manuscript. Most of my comments have been addressed; there are just a few points where further revision would be helpful:

In regard to attribution of adverse events, please indicate who made the attribution (e.g., investigators, participants, sponsors, or combinations) and whether they were blinded (it's not explicit whether "double blind" includes them)

Regarding the dose escalation arrows added to figure 1 - I realise this was in response to my suggestion of visual depiction of the dose escalation, but the implementation is confusing - because the figure is a patient flow diagram, use of solid lines suggests that the same patients received progressively higher doses (rather than the different doses being used in separate cohorts). I suggest depicting the dosing schedule separately, or at least changing the lines to dotted lines. (The description of the groups and dosing in the summary is now much clearer, thank you).

Please also describe the criteria for deciding to implement (or not) each dose escalation - was there a threshold for adverse events at which the trial would have stopped?

Reviewer #3 (Remarks to the Author):

In my opinion, the authors correctly address the main points reviewed.

Reviewer #4 (Remarks to the Author):

The manuscript is much improved. Thank you for your revision. I have two residual concerns. One new and one that remains to be addressed in a satisfactory way.

1. First, the new concern. In the revision, Aramchol has been added in the introduction as an agent in phase 3. This is not the case. While Aramchol has completed phase 2, it is NOT

currently in phase 3 development. Please remove this from the introduction.

2. Second, a residual concern and one that is quite important to align on. You are NOT studying NASH patients. The criteria you are using are too obtuse and do not justify appropriately the use of "NASH" in the title or throughout the paper. Please change this to NAFLD in the title and throughout the paper.

A minor note. Line 462, NAFLD is misspelled NALFD

Point-by-Point Responses: NCOMMS-22-49299B

Reviewer #1 comments to (some) responses

Reply to Reviewer #1:

Dear Editor,

As a statistician, I mainly focused on the quantitative aspects of the manuscript.

Comments/questions:

1/ 'Multi-dose escalation' & 'ascending doses':

These concepts, used on lines 127, 155, 232, 475, 427, 449 and 551, usually refer to adaptive trials in which allocation to the dose of the cth cohort depends on data collected on cohorts 1 to c- 1 (like seemingly considered in the phase I trial mentioned on line 115). It doesn't seem to be the case this phase Ib/IIa. Please clarify or prefer 'multiple-dose trial' without mentioning 'escalation' or 'ascending'.

Response: We are grateful to you for providing us with thoughtful comments and suggestions, which have helped us in improving our manuscript. With respect to the issues on the quantitative aspects, we have made the following amendments or explanations.

I thank the authors for sharing their raw data with reviewers during this revision process: this is **very much appreciated**.

Actually, this is an ascending dose study and we have made it clearer in Figure 1. In addition, we have checked the entire manuscript and made revisions as well.

I disagree. 'Multi-dose escalation' or 'ascending doses' typically implies the use of a design in which a **stepwise approach** starting with a low dose and gradually increasing the dosage level (according to some rules) is used. This is not the case here where you consider different doses in a parallel design. I therefore stick to my suggestion of “multi-dose trial” (which implies a set of increasing doses when ranked by dose levels). This position is shared by a few colleagues working on dose-finding.

Response: We agree with your additional comment and have incorporated the suggested change of "multiple-dose trail" in the revised manuscript.

3/ 'Sample size':

Phase Ib/IIa trials typically consider small sample sizes. Could the authors still mention why n=36 (Figure 1) was preferred to n=24 or n=45 for example, or, if they started with N=95 (Figure 1), why 95 patients were initially considered and not 80 or 120. Does it correspond to a number of patients per unit of time in the centres of interest?

Response: After reading your comments, we have reviewed the patient enrollment and allocation. A total of 95 patients were screened, and the number represented an actual number of screened patients, which cannot be estimated. The number (n=36) represented the final eligible patients for enrollment. The determination of 36 cases is to consider the first patient trial, and to ensure statistical needs as much as possible in the event of subject detachment.

Thanks. The question remains unanswered though: why 95 and not, say, 80 or 120?

Response: Thank you for your question, and we would like to provide the following explanation to clarify any confusion. Prior to randomization, screening examinations were conducted on the subjects. Only those who met the inclusion criteria were deemed eligible for randomization. However, the specific number of individuals for screening was uncertain. For instance, if we aimed to include 12 subjects in a group, we initially identified 10 potential candidates. Subsequently, these 10 subjects underwent screening examinations, but ultimately only 3 met the criteria. As a result, we needed to screen additional subjects until we reached a sufficient number of 12 eligible participants. However, the exact number of individuals for screening remained uncertain. It could potentially be 20, 32, or even more.

4/ 'Statistical analysis':

Some (significant) rewriting of the statistical analysis may be of benefit to the reader. 4a) MEANS:

4a- 1/ lines 229 to 233: "LS means": Authors seems to be performing "tests of equality of means between groups and/or time points with Tukey's multiplicity correction" (this sounds much clearer 'LS means corrected by Tukey's method') depending on the analysis. From their description, it is unclear what analysis was actually performed. For example

* in Table 4: are the 'LSmean (95% CI)' and the LSmean difference (95% CI) corrected for multiplicity?

* in Figures 2 and 3: how are the confidence intervals defined and are there corrected for multiplicity?

Therefore, it may be useful to list all ways of getting estimates and confidence intervals (with and without confidence intervals) and to specify in which Tables and Figures they were used.

Response: Thank you for your comment. Analysis of Covariance was used to calculate the LSmean of efficacy indicators and the LSmean difference (95% CI) corrected by Tukey's method.

LSmean (95% CI) and the LSmean difference (95% CI) in Table 4, and the data presented in Figures 4 & 5 with Tukey's multiplicity correction. As you suggested, we have made an amendment to the sentence in the revised manuscript

I am again **very grateful** to the authors to have shared their raw data as it allowed to just check if I can reproduce randomly chosen results.

In the section efficacy, I can reproduce the results of lines 339-340 (relative decrease in ALT levels) and 346, but not the absolute decrease levels of line 338 which makes me wonder how they were defined. For each patient, I defined the difference between ALT at day 29 minus ALT at baseline and modelled this difference as a function of dose level (ie, outcome = difference in ALT level, dose level as only predictor). Did you use a different model?

Response: Thank you for your comment. Line 338: The last version, the values in parentheses represent the LS mean and 95% confidence interval and not use different model. However, after model checking, ALT did not conform to normality and homogeneity of variance, so Kruskal Wallis test was used for indicators that to further verify the statistical differences (e.g, ALT,AST,GGT,FAST score and ALT in patients with a $BMI \geq 28$ kg/m²) and Tukey's test was used to correct for multiplicity. Draw conclusions based on statistics and revise corresponding descriptions.

I did the same for AST and can reproduce the results of lines 357-358 for all conditions but 50 mg BID (manuscript indicates 28.76% while I get 31.7%) and don't seem able to reproduce the results related to the absolute AST difference of lines 361-362, likely for the same reason as above.

Response: Thank you very much for bringing this issue to our attention. After carefully reviewing your comment, we thoroughly examined the original data. It was determined that there was an oversight during the data organization process, resulting in the incorrect baseline AST value for subject number 20301 in the 50 BID group. The correct value should be 57.9, not 98. We would like to assure you that the data presented in the manuscript is accurate.

4a-3/ model check

In small sample, the coverage of 95% confidence intervals based on the linear model *strongly* depends on the respect of linear model assumption (normality, homoscedasticity). A statement claiming that model checks were performed for each outcome and showed respect of the normality and homoscedasticity assumptions on the chosen scale (original or log scale, for example) would be useful.

Response: Thank you for your comment. Considering that, in this exploratory small-scale study of short duration, model check was not used.

From a statistician point of view, your answer is golden. You draw conclusions based on the use of statistical techniques. These conclusions may not be valid if the related assumptions are not met **especially** when the sample size is too small to rely on the central limit theorem, i.e., **especially** for small-scale studies.

We appreciate your suggestion, and will keep it in mind in our future study.

I strongly suggest that you perform model checks, at least for all analyses leading to statistical significance (ie, focusing on potential false positive). For efficacy outcomes, this means checking that the conditional normality and homoscedasticity assumptions are met so that you can claim that “*model checks showed a good fit of the model to the data*” in the manuscript. This will reassure this reviewer and a few of your readers. **Model check is not an option with small sample sizes.**

Response: As you suggested, we conducted model check on statistically significant indicators to evaluate the normality and homogeneity of variance of the raw data. After validation, analysis of covariance was conducted for indicators that conformed to normality($P>0.05$) and homogeneity of variance($P>0.1$), Kruskal Wallis test was used for indicators that did not conform to normality and homogeneity of variance to further verify the statistical differences (e.g, ALT,AST,GGT,FAST score and ALT in patients with a $BMI\geq 28$ kg/m²). After Kruskal Wallis test, if there is statistical significance($P>0.05$), we will continue to use Tukey ‘s test to correct for multiplicity between groups. Ultimately, a more reliable conclusion can be drawn. We can see from the current conclusion that there are still statistical differences in the results except for the change in ALT when $BMI\geq 28$ kg/m² compared the conclusion by ANCOVA.

5/ 'Demographics' and 'Randomisation':

Authors claim, on line 251 to 253, that demographic characteristics of the study subjects at baseline were comparable between treatment and control. Some may dispute this by pointing out that the proportion of females is (significantly, according to a Fisher test) larger in the control group (4/9) than in the treatment arms (3/27). Some may also claim that, given the statistics provided in Table 1, the patients of the control group are often older than patients in the treatment arms. It may be useful to explain how age and sex were randomised, to possibly consider a sensitivity analyses controlling for such imbalances (we recognise this may be difficult given the small sample size) and to display age (y-axis) per group (x-axis) in a plot (1 point per participant, coloured coded by sex) as supplemental material. If authors believe that such unbalances are not problematic as sex and age are not believed to be confounders, they should say so.

Response: Thank you for your thoughtful comments. We reviewed the original data of the proportion of females in the control group (4/9), which is larger than that in the treatment arms (3/27), and the patients in the control group are often older than those patients in the treatment arms.

Thanks. I think it is important to mention the fact that the randomisation led to some level of imbalance in the sex ratio and average age between conditions.

Thanks to the authors sharing their raw data, I could note that the average age is indeed significantly higher in the control group (the higher the dose, the lower the age). This may lead to a potential confounding when authors detect a relationship between an outcome and the dose level: **is it due to the dose difference or to the age or sex?**

We believe that such unbalances are not problematic on the basis of our observation in the phase Ia clinical trial (Zhu et al., 2021), in which the healthy subjects were enrolled at the male-to-female ratio of 1:1 and this is a randomized placebo-controlled ascending dose study, not a parallel trial, with a small sample size in each group. Age stratification was not conducted during randomization. In light of your comment, we have rewritten the statement of demographic characteristics of the study subjects at baseline in the revised manuscript.

I am afraid I don't think that this argument (mentioned on lines 279-281) is very convincing. If authors want to claim that their conclusions are valid while facing some level of imbalance imbalance in the sex ratio and average age between conditions, they should explain that sex and age have a marginal effect on the outcome of interest given the dose (or an effect going on the opposite direction of the dose effect).

Again, authors may be willing to check that their conclusions hold while controlling for age and/or sex which seems pretty straightforward in the ANCOVA framework they explained using above. They could then either [i] mention in the manuscript that "Sensitivity analyses controlling for age led to the same conclusions" which would reassure this reviewer and some readers or [ii] explain that causality is hard to define due to potential confounding in specific cases.

Response: Response: Thank you for your additional comment, I am so sorry that I did not provide a clear description of the first response to this comment. Firstly, in the phase Ia clinical trial (Zhu et al.,2021) of ZSP1601, we conducted a study on gender differences among healthy subjects(male-to-female ratio of 1:1) and did not find any gender differences in PK and safety results. Additionally, this study is a short-term study with a small sample size, in which each dose group consisted of 12 subjects randomized in a 3:1 ratio, at the same time, considering that incidence and prevalence of NAFLD are significantly higher among men than among women (Riazi K et al.,2022)and it was also reported in Asian (Le MH, et al.,2023), we did not specify the proportion of men and women in each dose group in the study design. we have no requirements for the male to female ratio of each dose group in the study design.

Regarding the issue of age, this imbalance is caused by randomness. However, from the clinical perspective of NAFLD in this study, age is not the main factor affecting efficacy, so this imbalance can be considered meaningless. At the same time, referring to clinical trials of drugs for treating NAFLD, there is almost no stratification of age and performing sensitivity analysis, especially with the a small sample size. Despite all this, we acknowledge that the age and gender imbalance is one of the limitations of this study and has been included in the Discussion section.

"There was an imbalance in the distribution of age and sex among the study subjects at baseline between the ZSP1601 and placebo groups that was caused by randomization (Table 1). Males had significantly higher incidence and prevalence rate of NAFLD, consistent with literature reports [30-31]. "

Reference

30. Riazi K, Azhari H, Charette JH, Underwood FE, et al. The prevalence and incidence of NAFLD worldwide: a systematic review and meta-analysis. *Lancet Gastroenterol Hepatol.* 2022 Sep;7(9):851-861.

31. Le MH, Le DM, Baez TC, Wu Y, et al. Global incidence of non-alcoholic fatty liver disease: A systematic review and meta-analysis of 63 studies and 1,201,807 persons. *J Hepatol.* 2023 Aug;79(2):287-295.

13/ 'Figure 3':

* in the legend, you forgot to mention you also display GGT (C) and ALP (D).

* the confidence intervals overlap: maybe consider shifting them slightly on the x-axis to allow comparisons.

Response: Thank you for bringing this issue to our attention, and we have added the information about GGT (C) and ALP (D) in the legend for Figure 3.

With respect to the comment (the confidence intervals overlap), we reviewed the original data presented in Figure 3. In fact, the data illustrated Figure 3 were mean \pm standard deviation (SD) but not confidence intervals. After reading your comment, we realized the SD error bars overlapped, while it is hard to shift them on the x-axis because of sharing the same time point.

Basing the plot on an amended dataset I which very slightly shift the time points per condition (+/- 0.1, for example) may be useful to avoid such overlaps

Response: Thank you very much for your valuable suggestion, which has greatly helped us improve the quality of the Figure 3. Accordingly, we have made slight adjustments to the horizontal axis values in order to minimize overlap as much as possible.

14/ 'Figure 4':

* your sample size per group is small enough to show all the individual data points.

Please do so.

* remember to indicate in the stats section if multiplicity correction per outcome in this analysis. If so, probably have this reflected on lines 308 and 309 by preferring adj. p-value to p-value.

Response: We have presented all the individual data points in the revised Figure 4. As you suggested, we have indicated multiplicity correction for the efficacy indicators under the heading of Statistical analysis in Methods section.

Thanks. As above, basing the plot on an amended dataset I which you very slightly shift the points on the x-axis may be useful to avoid overlaps of points.

Response: Thank you very much for your valuable suggestion, which has greatly helped us improve the quality of the Figure 4. Accordingly, we have made slight adjustments in order to minimize overlap as much as possible.

Reviewer #2 (Remarks to the Author):

Reply to Reviewer #2:

The authors have put considerable work into revising the manuscript. Most of my comments have been addressed; there are just a few points where further revision would be helpful:

In regard to attribution of adverse events, please indicate who made the attribution (e.g., investigators, participants, sponsors, or combinations) and whether they were blinded (it's not explicit whether "double blind" includes them)

Response: We appreciate your acknowledgement of our efforts in revising the manuscript. Regarding the additional comment on the attribution of adverse events, it is indeed determined by the investigators. This study is a double-blinded trial, in which investigators, participants, sponsors, or any combination of them are unaware of allocation of the treatment drug.

Regarding the dose escalation arrows added to figure 1 - I realise this was in response to my suggestion of visual depiction of the dose escalation, but the implementation is confusing - because the figure is a patient flow diagram, use of solid lines suggests that the same patients received progressively higher doses (rather than the different doses being used in separate cohorts). I suggest depicting the dosing schedule separately, or at least changing the lines to dotted lines. (The description of the groups and dosing in the summary is now much clearer, thank you).

Response: As you suggested, we have changed the lines to dotted lines in the revised Figure 1.

Please also describe the criteria for deciding to implement (or not) each dose escalation - was there a threshold for adverse events at which the trial would have stopped?

Response: Thank you for your thoughtful comment and suggestion. Based on CTCAE 5.0, dose escalation will be halted if any of the following conditions are met: (1) More than half of the subjects experience grade 2 or higher drug-related adverse events, (2) More than a quarter of the subjects experience grade 3-4 drug-related adverse events, or (3) a drug-related serious adverse event (SAE) occurs. The next highest dose level was initiated after a 14-day safety observation of the previous by principal investigator and sponsor.

Reviewer #3 (Remarks to the Author):

Reply to Reviewer #3:

In my opinion, the authors correctly address the main points reviewed.

Response: We are pleased to learn that you are satisfied with it. Additionally, we are grateful for your recognition of our work.

Reviewer #4 (Remarks to the Author):

Reply to Reviewer #4:

The manuscript is much improved. Thank you for your revision. I have two residual concerns. One new and one that remains to be addressed in a satisfactory way.

1. First, the new concern. In the revision, Aramchol has been added in the introduction as an agent in phase 3. This is not the case. While Aramchol has completed phase 2, it is NOT currently in phase 3 development. Please remove this from the introduction.

Response: Thank you very much for providing us with helpful information. As per your suggestion, we have removed it from the Introduction section.

2. Second, a residual concern and one that is quite important to align on. You are NOT studying NASH patients. The criteria you are using are to obtuse and do not justify appropriately the use of "NASH" in the title or throughout the paper. Please change this to NAFLD in the title and throughout the paper.

Response: We agree with your comment and have made the change to NAFLD in the title and throughout the manuscript.

A minor note. Line 462, NAFLD is misspelled NALFD.

Response: Thank you for bringing this typographic error to our attention. It has been corrected.

REVIEWERS' COMMENTS

Reviewer #1 (Remarks to the Author):

I agree with other reviewers that the manuscript is much improved.

I am happy with the changes made by the authors.

[Editorial Note: Reviewer name redacted.]